# High-throughput microcircuit analysis of individual human brains through next-generation multineuron patch-clamp

Yangfan Peng[1,2]*, Franz Xaver Mittermaier[1], Henrike Planert[1], Ulf Christoph Schneider[3], Henrik Alle[1], Jörg Rolf Paul Geiger[1]*

[1]Institute of Neurophysiology, Charité – Universitätsmedizin Berlin, Berlin, Germany; [2]Department of Neurology, Charité – Universitätsmedizin Berlin, Berlin, Germany; [3]Department of Neurosurgery, Charité – Universitätsmedizin Berlin, Berlin, Germany

**Abstract** Comparing neuronal microcircuits across different brain regions, species and individuals can reveal common and divergent principles of network computation. Simultaneous patch-clamp recordings from multiple neurons offer the highest temporal and subthreshold resolution to analyse local synaptic connectivity. However, its establishment is technically complex and the experimental performance is limited by high failure rates, long experimental times and small sample sizes. We introduce an in vitro multipatch setup with an automated pipette pressure and cleaning system facilitating recordings of up to 10 neurons simultaneously and sequential patching of additional neurons. We present hardware and software solutions that increase the usability, speed and data throughput of multipatch experiments which allowed probing of 150 synaptic connections between 17 neurons in one human cortical slice and screening of over 600 connections in tissue from a single patient. This method will facilitate the systematic analysis of microcircuits and allow unprecedented assessment of inter-individual variability.

*For correspondence:
yangfan.peng@charite.de (YP);
joerg.geiger@charite.de (JöRPG)

**Competing interests:** The authors declare that no competing interests exist.

## Introduction

Neuronal microcircuits represent the backbone for computations of the brain. While it is well established that feedforward information flow between the cortical layers is rather stereotypical (*Douglas and Martin, 2004*), intralaminar connectivity between different classes of neurons is more complicated and heterogeneous across brain regions (*Jiang et al., 2015*; *Peng et al., 2017*; *Steiner et al., 2019*; *Song et al., 2005*). Furthermore, it remains as yet unclear how observed microcircuit topologies generalize across species, or whether inter-individual variability can be observed at the level of neuronal microcircuits.

Cortical resections from epilepsy or tumor patients represent a unique opportunity to analyze the human cortical microcircuit. While specific cellular and synaptic properties in the human central nervous system have been described (*Kalmbach et al., 2018*; *Molnár et al., 2016*; *Szegedi et al., 2016*), distinctive features at the microcircuit level have yet to be resolved (*Seeman et al., 2018*). Availability of human brain tissue is scarce, making it, at present, necessary to pool data across patients. This is problematic considering the high diversity between patients with respect to different environmental and genetic background, medical history and medication. Therefore, it is essential that we develop more efficient ways to generate large sample sizes from single patients to assess inter-individual variability. This could then lead to the identification of invariant parameters pointing toward common principles of the human cortex, while parameters with high inter-individual variability should be subject to further investigation. Apart from that, in non-human studies, increasing the data yield from individual animals would efficiently promote the ethical principle to replace, reduce and refine animal experiments (*Russell et al., 1959*).

To dissect microcircuits, multiple whole-cell patch-clamp recordings (multipatch) still represent the gold standard method compared to various other approaches (*Hochbaum et al., 2014*; *Packer et al., 2015*). This method can reliably detect unitary excitatory and inhibitory synaptic connections due to its sub-millisecond and subthreshold resolution (*Geiger et al., 1997*). In addition, it allows for detailed electrophysiological and morphological characterization of recorded neurons and can be used across species without the need for genetic modifications in contrast to optogenetics or calcium imaging (*Markram et al., 2004*). These conditions enable the investigation of connection probabilities, synaptic strength, plasticity and higher order network statistics incorporating distance-dependencies (*Song et al., 2005*; *Thomson and Lamy, 2007*). Increasing the number of simultaneously recorded neurons can increase the number of probed synaptic connections considerably and generate larger sample sizes from fewer experiments.

Multipatch setups increase the complexity and time of experiments, necessitating automation. Various groups were able to increase the number of simultaneous recordings (*Guzman et al., 2016*; *Jiang et al., 2015*; *Peng et al., 2017*; *Perin et al., 2011*; *Perin and Markram, 2013*; *Wang et al., 2015*), but the operation of multiple manipulators is challenging. To address this, several approaches have been reported that automate the patch-clamp process, utilizing automated pressure control systems and algorithms for manipulator movements guided by visual or electrical signals (*Kodandaramaiah et al., 2018*; *Kodandaramaiah et al., 2016*; *Kodandaramaiah et al., 2012*; *Perin and Markram, 2013*; *Wu et al., 2016*; *Kolb et al., 2019*). However, while simultaneous recordings of up to four neurons are becoming increasingly popular, eight or more manipulators on one setup are currently used by only a few labs, missing out on the opportunity to maximize the potential of this method.

Cleaning pipettes for immediate reuse increases the size of recorded neuron clusters and enables a more complete view of the microcircuit. The maximum number of simultaneously recorded neurons is highly limited by the spatial constraints imposed by the manipulators. Furthermore, the success of a whole-cell recording depends on mechanical interference, deterioration of recording quality during prolonged experimental time and tissue quality. These factors are aggravated when the number of pipettes is increased. Cleaning pipettes for immediate reuse enables recording of additional neurons in one experimental session without manual replacement (*Kolb et al., 2016*). This technique has already been implemented for automated patch-clamp of single neurons in vivo (*Kodandaramaiah et al., 2018*) and in vitro (*Kolb et al., 2019*). Applying pipette cleaning to in-vitro multipatch setups has the potential to increase the number of pipettes with successful recordings and therefore the experimental yield. Sequential recordings from multiple cells using the same pipette would also overcome the limitation on maximum cluster size given by the number of manipulators in use. This will provide a more complete view of the microcircuit enabling the analysis of more complex network motifs and higher degree distributions (*Perin et al., 2011*; *Song et al., 2005*; *Vegué et al., 2017*).

We introduce an in vitro multipatch approach with an automated pipette pressure and cleaning system to record up to 10 neurons simultaneously. Here, we show that this approach and further optimizations increase the rate of complete clusters, decrease experimental time and enable sequential patching of additional neurons. We demonstrate that probing of up to 600 synaptic connections in human brain slices in one day is possible with two setups, allowing analysis of microcircuits at the level of individual patients.

## Methods

We first provide technical instructions for establishing a multipatch setup and how to address specific challenges when using up to 10 manipulators. We then describe an automated pressure device for multiple pipettes and how to implement a pipette cleaning protocol. Finally, we show that these technical improvements enable a high data yield allowing extensive connectivity analysis in slices of human cortical resections and higher degree analysis of individual cells far exceeding the capacity of conventional multipatch setups.

### Arrangement of hardware components

Expanding a setup from a single or dual patch-clamp configuration to one with more patch pipettes is primarily an issue of spatial arrangement of the micromanipulators. The upscaling of multipatch

setups has been accomplished with manipulator systems from different commercial manufacturers such as Luigs and Neumann (*Guzman et al., 2016*; *Jiang et al., 2015*; *Wang et al., 2015*; *Winterer et al., 2017*) and Scientifica (*Cossell et al., 2015*; *Peng et al., 2017*). Crucial to any multi-patch setup is the spatial arrangement of the manipulators which we found most easy to address by using a custom stage even though there are also commercial solutions available, such as from Luigs and Neumann and Scientifica. We arranged 10 Sensapex uMp-Micromanipulators on a setup with a Nikon Eclipse FN1 microscope using a custom-made stage which was fabricated by a commercial metal workshop (*Figure 1A–C*). We also successfully established a setup with 8 Scientifica PatchStar manipulators on a custom-made stage with a Scientifica SliceScope microscope (*Figure 1D*; *Peng et al., 2017*). Our custom stages consist of a 10 mm aluminium sheet for stability glued to a 1

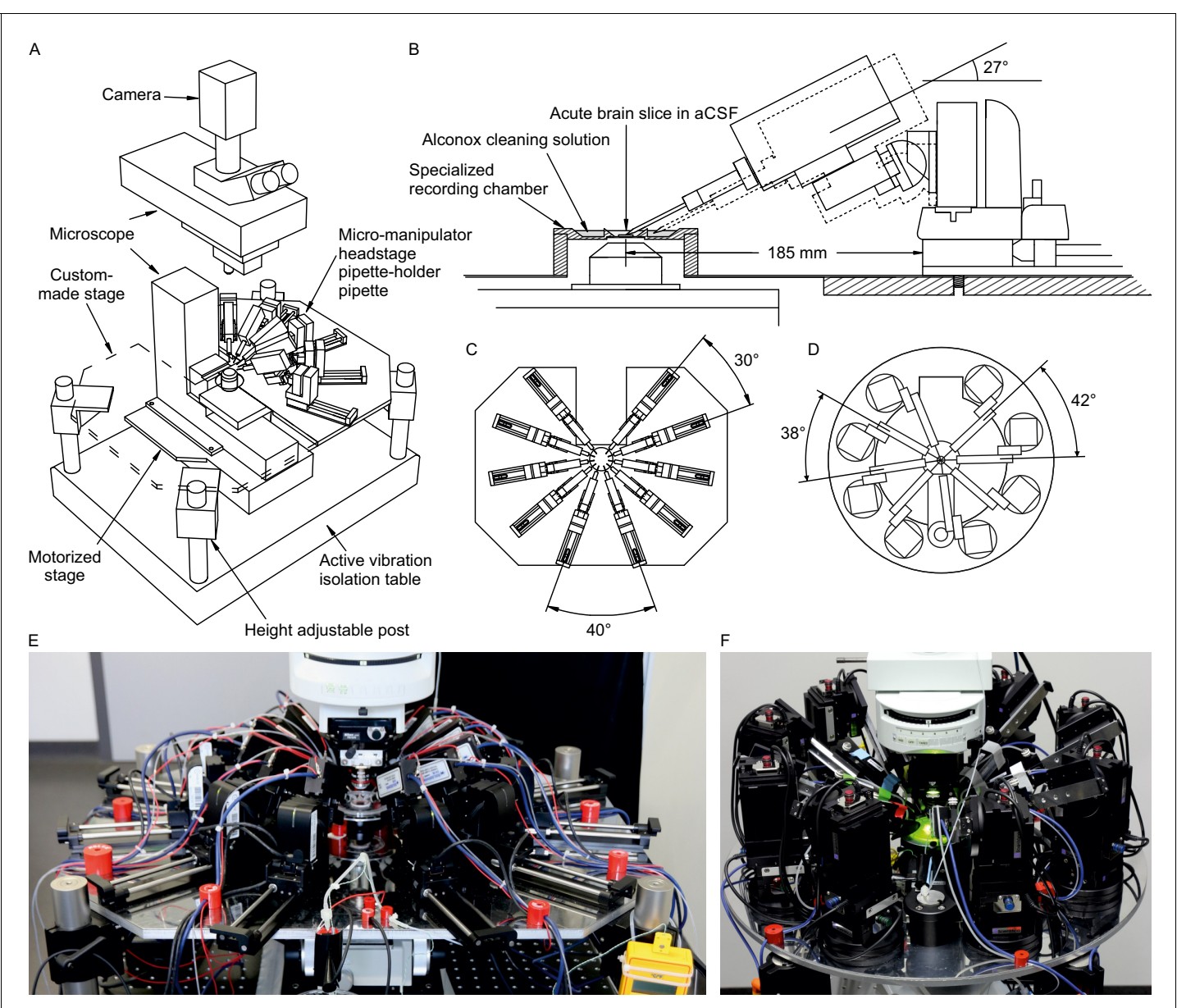

**Figure 1.** Multipatch setups. (**A**) Overview of essential components of a 10-manipulator setup with Sensapex micromanipulators. (**B**) Side view depicting the spatial arrangement of a manipulator including headstage and pipette relative to the recording chamber which is elevated above the condensor. The dashed outline shows the manipulator with pipette tip immersed in the cleaning solution. (**C**) Top view of 10 Sensapex manipulators with headstages and their angles to each other on the custom-made stage. (**D**) Top view of 8 Scientifica PatchStar manipulators with headstages and their angles to each other on a custom-made ring-shaped stage. (**E**) Photograph of the 10-manipulator setup. (**F**) Photograph of the eight-manipulator setup.

mm stainless steel sheet to create a ferromagnetic surface for cable routing using magnets. The stage is supported by platforms mounted onto poles, which enable flexible height adjustment. For exact dimensions of the stage, see *Appendix 3—figures 4* and *5*.

Another crucial aspect is the distance and angle of the pipettes to the recording chamber (*Figure 1B*). A shallow approach angle increases the range of motion in the x-axis which is important to reach both the centre of the recording well and the outside well containing the cleaning solution. However, the angle needs to be sufficiently steep to allow the pipette tip to reach into the slice in the z-axis while avoiding touching the wall of the centre well. We found 27° to be optimal for our custom-made recording chamber, which is positioned on an elevated platform. To increase stability, the distance of the manipulator to the recording chamber should be minimized. At the same time, sufficient space between headstages must be maintained to allow the necessary range of motion.

Digitizers with sufficient recording and stimulation channels in one device combined with the respective software solution offer a higher experimental flexibility and easier implementation than separate devices. On the 10-manipulator setup, five MultiClamp 700B amplifiers (Molecular Devices) were connected to one digitizer (Power1401-3A with Signal extension from Cambridge Electronic Design) enabling simultaneous recording of 10 channels within the corresponding commercial software Signal (*Table 1*). Amplifier settings were controlled manually through multiple instances of the MultiClamp 700B Commander software (Molecular Devices). Since there is currently no commercial hardware solution that incorporates more than eight digital-to-analogue converter (DAC) output channels with the same number of analogue input channels in one device, we needed to increase the number of DAC output channels from 8 to 10 on the 10-manipulator setup. We achieved this by a simple routing device that can switch between channels enabling four headstages to receive stimulation from only two DAC outputs (for details and limitations of the routing device see *Appendix 3— figure 11*).

## Pressure control system

We adapted the use of solenoid valves and pressure regulators to our in vitro multipatch approach. Throughout the process of patching a cell, variable pressure levels are required. Conventionally, this is achieved with pressurized chambers, a three-way valve and a mouthpiece or a syringe. Since control of pressure in multiple pipettes via manual setting of valves becomes complex and error-prone, an automated pressure system is of advantage. A previous report has demonstrated an electrically controlled pressure system for a multipatch setup (*Perin and Markram, 2013*). Other groups have also implemented automatic pressure devices together with pipette movement algorithms for automated in vivo patch-clamp (*Desai et al., 2015*; *Kodandaramaiah et al., 2016*; *Kodandaramaiah et al., 2012*). The *Autopatcher* by Kodandaramaiah et al. has also been developed for multi-neuron in vivo recordings with four pipettes (*Kodandaramaiah et al., 2018*). We adapted the described automatic pressure system to our setup and scaled it up to 10 pipettes, with different components that are cost-efficient and widely available. It also enables the simultaneous application of multiple pressure levels to individual pipettes. A detailed component list with price estimates and construction manual with illustrated step-by-step instructions for the automated pressure system can be found in appendix 1. It can be built within a week with basic electrical equipment and does not require extensive technical skills. We believe that the easy-to-build modular construction approach is useful for labs that want to implement this tool and adjust it to their needs.

Multiple pressure regulators were connected to a compressed air outlet to generate different adjustable pressure levels to the pipettes (*Figure 2A*). We set one pressure regulator at 20 mbar (LOW) for continuous outflow of pipette solution during the idle state in the bath to prevent clogging of the tip. We apply 70 mbar (HIGH) for moving through the slice to approach the cell. Since HIGH pressure is only applied to the active pipette and will always be followed by the pressure of a mouth piece or syringe (PATCH), these two pressure levels share one output. In the PATCH mode, we wait for the formation of the gigaseal, apply light suction when needed and apply a stronger suction to break into the cell. We believe that this is a crucial part of the patching process which needs fine adjustment regarding the duration and strength of suction to achieve maximum success rate (up to 88% on six-manipulator setup, see results on performance below). Compared to this, reports on fully automated in-vitro patch-clamp algorithms have reported a relatively low success rate of 43% using a single pipette (*Wu and Chubykin, 2017*). After successful establishment of a whole-cell

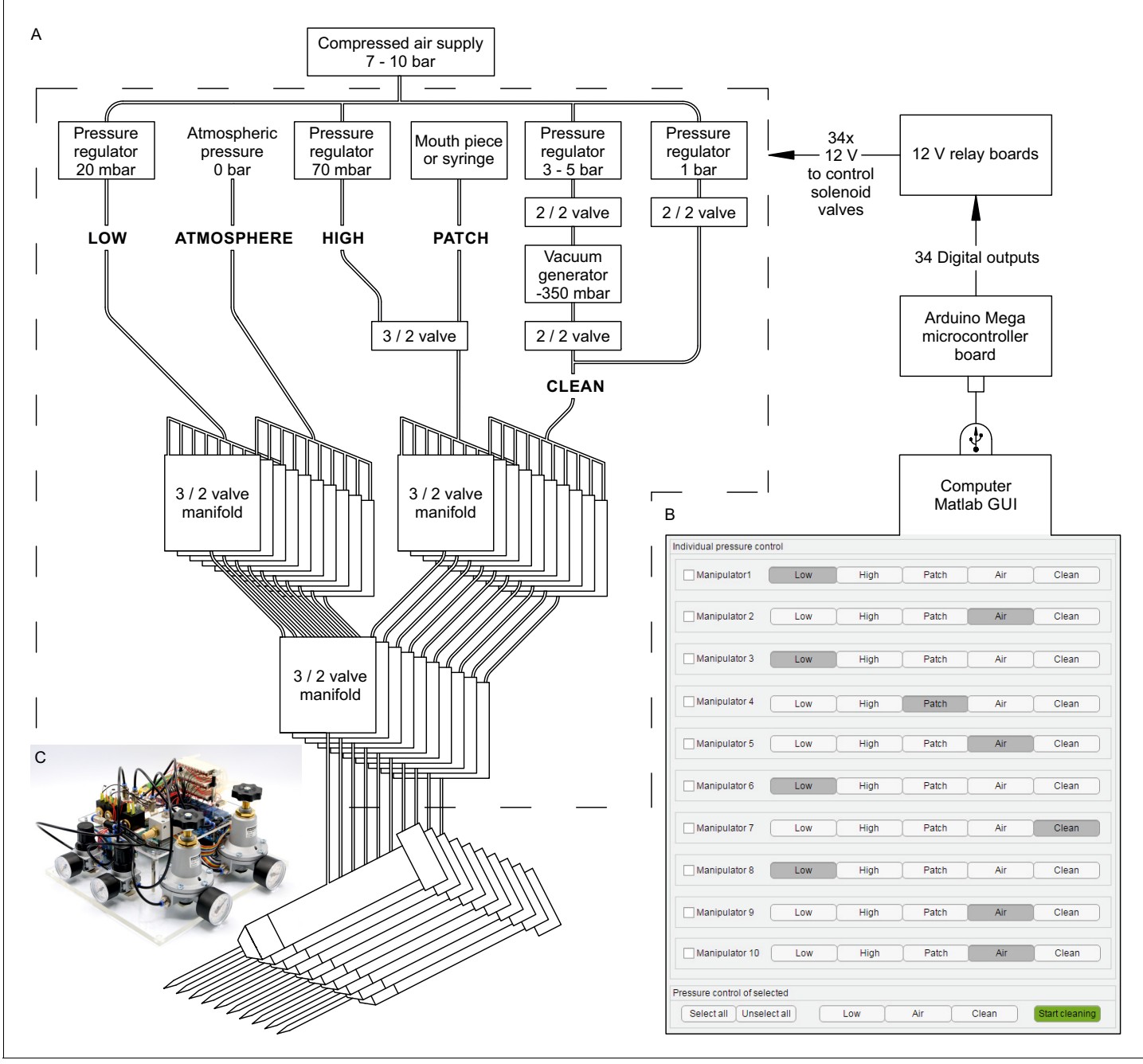

**Figure 2.** Automated pressure system. (**A**) Tubing scheme between the different pneumatic components. Top row depicts pressure regulators and their set pressure for each path. 2/2 valves are solenoid valves with two ports and two positions (open or closed). 3/2 valves are miniature solenoid valves with three ports and can switch between two positions connecting either inlet to one outlet. An illustrated step-by-step assembly guide of the pressure system can be found in Appendix 1. (**B**) Screenshot of Matlab GUI controlling the pressure system. It communicates with an Arduino microcontroller board which in turn has digital outputs connecting onto a relay board. For further details of wiring scheme see *Figure 2—figure supplement 1*. For more information on GUI see appendix 2. (**C**) Photograph of an assembled pressure system.

The online version of this article includes the following figure supplement(s) for figure 2:

**Figure supplement 1.** Wiring scheme of pressure system.

configuration, the pressure in the pipette will be switched to atmospheric pressure (ATMOSPHERE). The fifth pressure channel is implemented for the pipette cleaning process described below (CLEAN).

By arranging three miniature solenoid valves in a tree structure for each pipette, we are able to direct these different pressure levels to each pipette individually (*Figure 2A*). The solenoid valves are electrically operated using commercially available 12 V relays which are controlled by the digital outputs of an Arduino Mega microcontroller board. To control the pressure system through the Arduino board, we developed a graphical user interface in MATLAB (*Figure 2B*, further instructions are provided in Appendix 2, MATLAB code available ). This open-source and affordable pressure system is applicable to any other patch-clamp setup. It can reduce experimental errors, increase the experimental speed and help the experimenter to concentrate on the other tasks essential for successful operation of a multipatch setup.

## Cleaning system

We added a pipette cleaning protocol to the in vitro multipatch setup which enables immediate retry after a failed patch attempt and recording of additional neurons through sequential patching. A recent study has shown that dipping the tip of a patch pipette in a specific detergent solution (Alconox) and applying a sequence of pressure and suction could clean a pipette for immediate reuse (*Kolb et al., 2016*). In this study, more than 10 successful 'repatches' using the same patch pipette were possible while no change in the electrophysiological properties of the cells was observed. We implemented this protocol to our multipatch approach, because it was able to increase the success rate of establishing good recordings. To incorporate a well for the detergent solution, we constructed a custom recording chamber with an outer circular well which is separated to the inner recording well (*Figure 3A*, see appendix 3 for detailed construction designs and how wicking and overflow of the detergent is prevented). With the optimal angle and positioning of the manipulators (*Figure 1B*), all pipettes can access the brain slice and the cleaning solution without mechanical interference.

To clean the pipettes automatically, both the sequence of pipette pressure and manipulator movement need to be automated and coordinated. We programmed the movements of our Sensapex and Scientifica micromanipulators in Matlab and integrated them into the GUI of the pressure system. We recommend custom development of these manipulator movements due to the specific parameters of each setup. Example code and key aspects of manipulator programming can be found in the Github repository (link provided in appendix 2). Using the CLEAN channel, the automated pressure system is now able to apply positive and negative pressure as described in the protocol by *Kolb et al. (2016)*. We used a pressure regulator connected to the compressed air outlet to generate 1 bar for expulsion and another pressure regulator coupled with a vacuum ejector to generate −350 mbar for suction at the pipette tip (*Figure 2A*). The pressure regulators and the vacuum generator are connected to downstream solenoid valves which control and alternate the pressure.

Incorporating automated manipulator movements, pressure control and the cleaning protocol make the experimental steps easier and faster (*Video 1*). At the beginning of the experiment pipettes are sequentially moved into the recording well to their dedicated position in the field of view approximately 2 mm above the slice. Small manual adjustments under visual control are necessary for each pipette due to variation in pipette length. The pipette pressure is set to LOW (20 mbar) to avoid clogging of the tip (*Figure 3B*). To speed up this process and avoid collisions, we semi-automated it by developing a pipette finding algorithm which we discuss further below. After positioning of all pipettes around approximately 200 μm above the slice, cells are approached and patched sequentially under visual guidance. To prevent contamination of the pipette tip while moving through the slice, the HIGH pressure (70 mbar) is applied to the pipette (*Figure 3C*). When a dimple on the cell surface can be seen, the pressure is released by switching to the PATCH channel. Manual application of suction either through a mouth piece or manually through a syringe is then needed to obtain a good seal and to break through the membrane to establish the whole-cell configuration (*Figure 3D*). Pipettes with successfully patched cells are switched to atmospheric pressure (ATMOSPHERE, *Figure 3E*). A similar hybrid approach of manual and automatic pressure control has also been implemented in a previous multipatch study (*Seeman et al., 2018*). If the sealing process is unsuccessful or the whole-cell recording deteriorates, even multiple pipettes can now be cleaned simultaneously through an automated process. The cleaning process starts with the automated retraction of the pipettes to a position outside of the chamber above the outer well (*Figure 3F*). They are then lowered into the detergent solution (*Figure 3G*). After the pipette pressure is switched to the CLEAN channel, suction (−350 mbar, 1 s) and pressure (1 bar, 1 s) are applied for

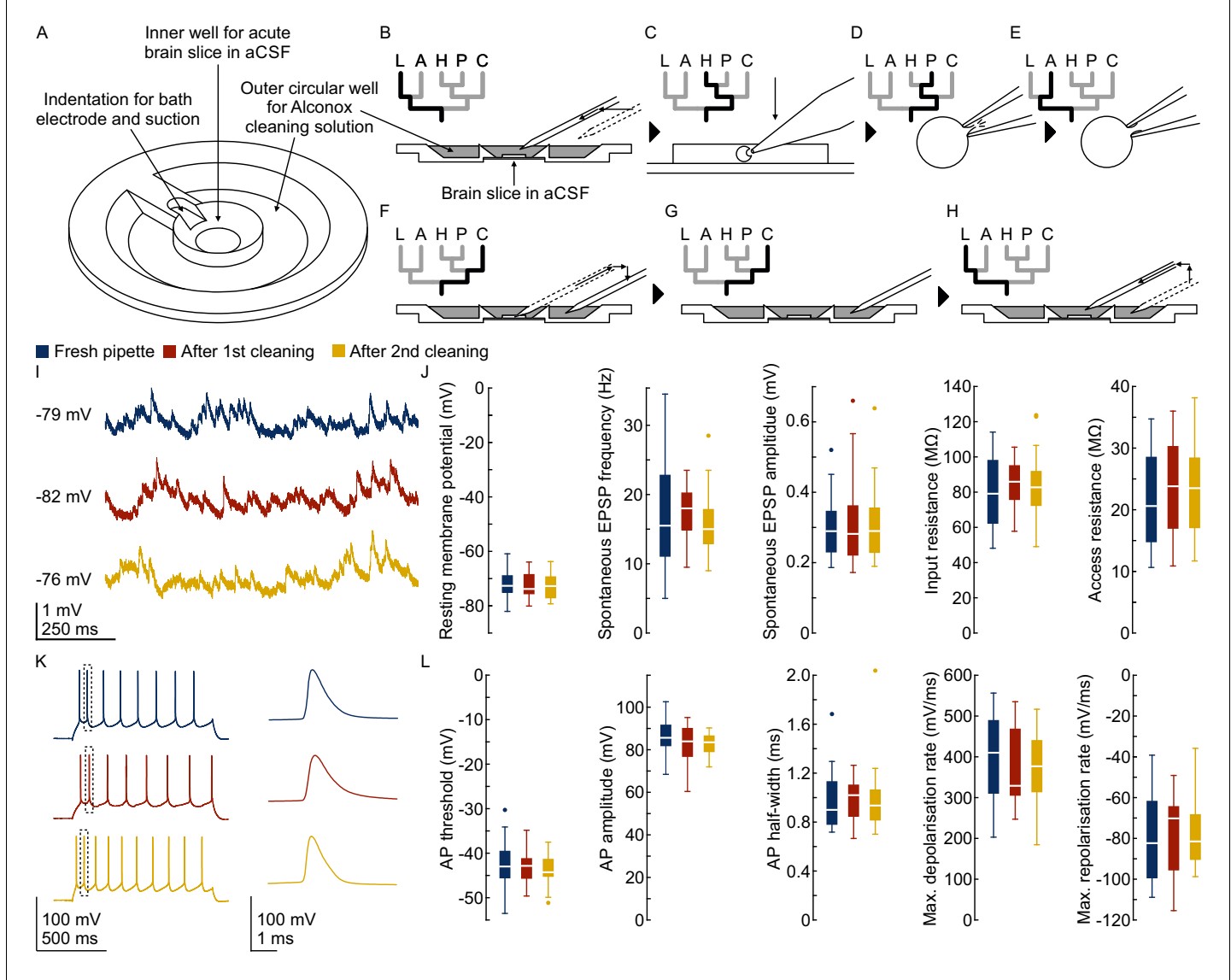

**Figure 3.** Pipette cleaning protocol. (**A**) Design of custom-made recording chamber with one centre well for the brain slice and recording solution and another circular outer well for the cleaning solution, see construction design in **Appendix 1—figure 1** and **2**. (**B–H**) Experimental steps for pipette cleaning. Tree diagrams on the top left depict the configuration of the pressure system with the following pressure levels: LOW, ATMOSPHERE, HIGH, PATCH, CLEAN. (**B**) Pipettes are moved into the recording solution with LOW pressure. (**C**) Cells are approached with HIGH pressure. (**D**) Formation of gigaseal and membrane rupture with pressures applied through the PATCH channel with either mouth piece or syringe. (**E**) Cells are kept at ATMOSPHERE pressure after successful patch. (**F**) Pipettes are moved into the cleaning solution and switched to CLEAN pressure. (**G**) Cleaning pressure sequence is applied in the outer well. (**H**) Pipettes are moved back into the recording solution above the slice and switched to LOW pressure. (**I**) Current clamp traces at resting membrane potential exhibiting spontaneous EPSPs recorded using the same pipette before (blue) and after cleaning (red and yellow). (**J**) Box plots of cellular, synaptic and recording properties of cells recorded with fresh (blue, n = 25, except n = 23 for synaptic parameters) and cleaned pipettes (red, n = 21 and yellow, n = 21). (**K**) Action potential firing patterns elicited through step current injection and enlarged action potential traces recorded on the same pipette before (blue) and after cleaning (red and yellow). (**L**) Box plots of action potential properties of cells recorded with fresh (blue, n = 25) and cleaned pipettes (red and yellow, n = 21). Further statistical information and analysis are shown in **Figure 3—source data 1**.

The online version of this article includes the following source data and figure supplement(s) for figure 3:

**Source data 1.** Statistical summary intrinsic parameters.
**Source data 2.** Statistical summary synaptic parameters.
**Figure supplement 1.** Fresh pipette vs. cleaned pipette.
**Figure supplement 2.** Fresh pipette vs. repatch of same cell with cleaned pipette.
**Figure supplement 3.** Fresh pipette vs. repatch with fresh pipette.

*Figure 3 continued on next page*

*Figure 3 continued*

**Figure supplement 4.** ACSF before vs. after rinsing during cleaning.
**Figure supplement 5.** Synaptic amplitudes recorded in ACSF before vs. after rinsing during cleaning.

five cycles, followed by a long expulsion sequence (1 bar, 10 s). Please note that the expelled volume is minimal relative to the total volume inside the pipette. For the next step, the pipettes are moved to the outer rim of the recording well into the recording solution for a final expulsion sequence (1 bar, 10 s) and are then moved to the initial position above the slice while the pressure is set to LOW (*Figure 3H*).

The final expulsion sequence does not necessarily require additional wells containing aCSF. To ensure access of all pipettes to the cleaning solution and due to limited range of motion of the manipulators, we omitted the second well that was introduced in the original protocol for the final expelling sequence (*Kolb et al., 2016*). Thus, in our approach the final expelling sequence might introduce traces of Alconox solution into the extracellular recording solution. Alconox contains 10–20% sodium linear alkylbenzene sulfonate (LAS), which affects Glycin-, GABA$_A$- and GluR6-mediated currents at concentrations around 0.001% (*Machu et al., 1998*). Since it has already been shown that no significant amount of LAS could be detected in the pipette solution after the cleaning cycles (*Kolb et al., 2016*), the remaining potential source of contamination is residual detergent adhering to the outside surface of the pipette tip. A volume of at least 25 µl Alconox would be needed to reach the critical LAS concentration of 0.001% in the extracellular solution, assuming a recording bath volume of 1 ml and a LAS concentration of 20% in the 2% Alconox solution. We estimate the adherent Alconox solution on a pipette tip to be less than 0.2 µl, considering that the total volume of intracellular solution inside a pipette is usually less than 15 µl. We thus believe that this high degree of dilution and the continuous flow of extracellular solution at more than 2 ml/min prevents impact on the synaptic and electrophysiological properties.

To test possible effects of pipette cleaning solution on cellular and synaptic properties, we performed dedicated experiments in rat brain slices (see method section for experimental details and exclusion criteria). We compared the cellular properties (resting membrane potential and input resistance), synaptic properties (spontaneous EPSP frequency and amplitude), action potential kinetics (threshold, amplitude, half-width, depolarisation and repolarisation rate) and recording quality (access resistance) of 67 rat pyramidal neurons before (n = 25), and after cleaning (n = 21 after first cleaning and n = 21 after second cleaning, *Figure 3I–L*). While we observed some variability between different recordings, we found no evidence for a systematic change in cellular or synaptic physiology (p>0.05 in all comparisons using repeated measures ANOVA, *Figure 3—source data 1*). The mean relative difference for all parameters were between 0% and 10%. We also calculated the 90% confidence intervals of the relative mean difference which indicates the boundaries of statistically significant equivalence in these parameters (within ±20% for cellular and action potential properties, within ±25% for spontaneous EPSP properties, *Figure 3—source data 1*).

We further assessed the effect of pipette cleaning in human brain slices. We first compared intrinsic properties of human neurons patched with fresh (n = 24) and cleaned pipettes (n = 9, *Figure 3—figure supplement 1*). The mean relative difference for all parameters were within 13% (p>0.05, *Figure 3—source data 1*). We also repatched the same neurons with the same, but cleaned pipette (n = 9, *Figure 3—figure supplement 2*) or another fresh pipette (n = 5, *Figure 3—figure supplement 3*). The mean relative differences in these conditions were within 14% (p>0.05), except for a 25% decrease in input resistance (p=0.01) and 20% decrease in access resistance (p=0.36) in cells repatched with a cleaned pipette. Multiple factors could contribute to the variability in these neurons, such as the effect of repatching itself or time passed. These cleaning-independent changes are reflected in the variability also found in neurons repatched with fresh pipettes (*Figure 3—figure supplement 3*).

As we have shown that the pipette itself has no systematic effect on the intrinsic properties of human neurons, we have also addressed the possibility that the extracellular solution, in which the pipettes were rinsed, could have an effect. We have therefore patched clusters of neurons (n = 19) and compared their properties (*Figure 3—figure supplement 4*) and synaptic connections (n = 7, *Figure 3—figure supplement 5*) before and after the cleaning of other pipettes to simulate changes

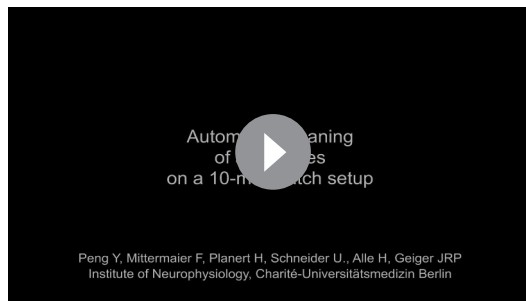 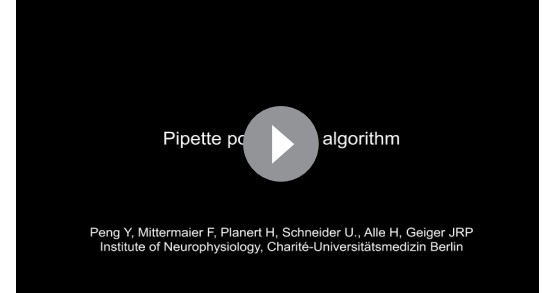

**Video 1.** Automated cleaning of all pipettes on a 10-multipatch setup. The main image depicts a front view of the 10-manipulator setup with all manipulators subject to automated movements during a pipette cleaning sequence. The insert on the bottom left corner shows a close-up view of the pipettes and the recording chamber. For better overview, the two front manipulators were moved aside. The insert on the bottom right corner shows the DIC microscope view with a 4x objective. At the end of the sequence, all pipettes are out of focus, due to different final z-positions. The time stamp in the middle shows the elapsed time in seconds. Roughly 20 s of alternating pressure sequence was cut (second 13 to 32). The last 8 s sequence shows the pressure system while switching between pressure levels. The clicking noise is generated by the relais switches.
https://elifesciences.org/articles/48178#video1

**Video 2.** Pipette positioning algorithm. This video shows the pipette movements during the positioning phase under the DIC microscope. The first sequence is in a 5x time lapse with the 4x objective. Pipettes are moved in and out automatically, note the small manual adjustments in between. The second sequence is played in real time and shows all pipettes moving to their adjusted target position underneath the 40x objective.
https://elifesciences.org/articles/48178#video2

in the extracellular solution after the rinsing. We found that the resting membrane potential and action potential kinetics remained very stable (mean relative difference within 2%, $p>0.05$) while the input and access resistance increased (input resistance 23%, $p<0.001$; access resistance 24%, $p=0.26$). We further did not see a specific trend in the postsynaptic amplitudes between these two conditions which showed both increase and decrease ($n = 7$, mean relative change within −25% to +26%, *Figure 3—figure supplement 5* and *Figure 3—source data 2*).

Overall, these results in rat and human neurons strongly suggest that our cleaning procedure does not alter the physiological properties in a systematic way and that it is therefore suitable to analyse synaptic connectivity. However, we did see that the input resistance decreased in cells repatched with a cleaned pipette and increased in cells recorded in ACSF after rinsing. While we did not detect these differences on the population level and despite the fact that repatching, recording time and neuronal variability could affect these parameters, we cannot definitely exclude an effect of pipette cleaning in these cases. We therefore emphasize that these validation experiments are limited to our setup and research questions and that any implementation of our cleaning procedure by others should be thoroughly tested for the specific experimental setting, especially when effects on the level of ion channels or receptors are to be studied. As the amount and possible effect of adhering detergent on the pipette can depend on multiple factors, we recommend fine-tuning of bath and detergent volumes in the chambers, ACSF flow rate, time and pressure for the second expulsion sequence and position of pipettes during rinsing combined with own control experiments.

The entire cleaning process takes approximately 1 minute and can be executed for multiple pipettes at the same time. Since mechanical interference is minimal, pipettes failing to establish a good recording can be subject to immediate cleaning in parallel with the patch attempt of another pipette. This greatly increases the success rate of obtaining recordings and thus the yield of a single experiment. Please note, however, that we do not recommend recording traces during ongoing patch attempts due to reasons we discuss further below. After recording of a full cluster, multiple pipettes can be cleaned, and additional cells can be patched and included in the analysis. We also believe that the possibility to repetitively perform patch attempts without manual replacement of the pipettes increases the speed of learning and performance of novel experimentalists, thus lowering the barrier to establish multipatch setups.

## Preparation of pipettes

With increasing pipette numbers, preparations take up significant experimental time which we addressed by additional optimizations such as a multi-pipette filling device and semi-automated positioning of pipettes. Before the pipettes are positioned above the slice and ready for patching, multiple time-consuming preparatory steps are necessary, such as pipette pulling, filling and positioning. We therefore designed a device which can hold and fill multiple pipettes at the same time (see appendix 3 for construction designs). It uses the prefill approach to suck intracellular solution through the tip. Backfilling using a syringe might still be necessary when prefilling is not sufficient. This approach reduced the time to fill and mount all pipettes to 11–13 min, while the time during prefilling can be used for other preparations (*Figure 4C*). We also optimized the procedure to find and position the pipettes in a new region of interest (*Video 2*). Manipulator coordinates were matched to the microscope coordinates using a rotation matrix and anchored to a common reference point (*Perin and Markram, 2013*). A semi-automated algorithm determines optimal positions of each pipette and all pipettes are moved to roughly 200 µm above the slice. The specific steps are explained in detail in *Appendix 2—figures 2 to 9*. This semi-automated approach reduces the risk of breaking pipette tips and the time needed to complete pipette positioning to 7–9 min (*Figure 4C*).

## Results

## Performances achievable with multipatch setups

When more neurons (n) are recorded simultaneously, the number of tested synaptic connections (c) scales according to $c = n \times (n - 1)$ (*Figure 4A*). However, increasing the number of pipettes lowers the rate of obtaining a full cluster with successful recordings on all pipettes, even for skilled experimenters. This is not only due to more patch attempts, but also due to the increased risk of losing established recordings through mechanical interference or deterioration of recording quality during the prolonged total time of approaching and patching cells. Generally, to minimize the distortion of the tissue during movement of the pipette through the slice, we recommend to first patch the cells furthest from the slice surface and to approach the target cell along the longitudinal axis of the pipette. In rare cases, slight readjustments of pipettes between patch attempts can be necessary. To provide an estimate for the performance of different setup configurations, we analyzed our multipatch experiments on rodent slices from previous and ongoing studies without pipette cleaning (*Böhm et al., 2015*; *Peng et al., 2017*). Success rate of patch-clamp recordings can vary greatly depending on tissue health, species, brain region and cell type. For our performance analysis, we focused on experiments in the mouse subiculum, rat presubiculum and rat motor cortex to present an applicable use case for labs working with rodent slices.

On a setup with six manipulators, we obtained on average 5.3 ± 0.7 successful recordings per cluster with an average of 23 ± 7 tested connections (mean ± standard deviation, n = 30). This corresponds to an average success rate of 88 ± 12% which is calculated as the ratio of successfully recorded cells to the maximum number of pipettes available per cluster (*Figure 4B*). We refrained from manual replacement of failed pipettes due to the risk of losing the other recordings. After scaling up to eight pipettes, we could record from 6.8 ± 1 cells on average (success rate 85 ± 13%) which increased the mean number of tested connections to 41 ± 13 (n = 33). On our setup with 10 pipettes, we recorded on average from 7.9 ± 1.1 neurons with 56 ± 17 tested connections which represents a further drop of the success rate to 79 ± 11% (n = 10). Concurrently, the total experimental time needed before the start of recording the electrophysiological properties of the neurons rose from 36.6 ± 2.3 min (eight pipettes) to 46.1 ± 4.6 min (10 pipettes), mostly due to increasing time needed to approach and patch the cells (16.8 ± 2.7 min vs 25.8 ± 2.2 min, *Figure 4D*). Considering this drop in success rate and increased experimental time, the upscaling of a conventional multipatch setup will become impractical at a certain point.

Recording neurons simultaneously to ongoing patch attempts is hindered by several practical limitations. The outflow of high-potassium intracellular solution depolarises the other cells during the patch attempt which would affect ongoing nearby recordings. Furthermore, to monitor the gigaseal formation, we would need to set the respective channel to voltage clamp mode while synaptic connections are recorded in the current clamp mode, both with different stimulation protocols. To

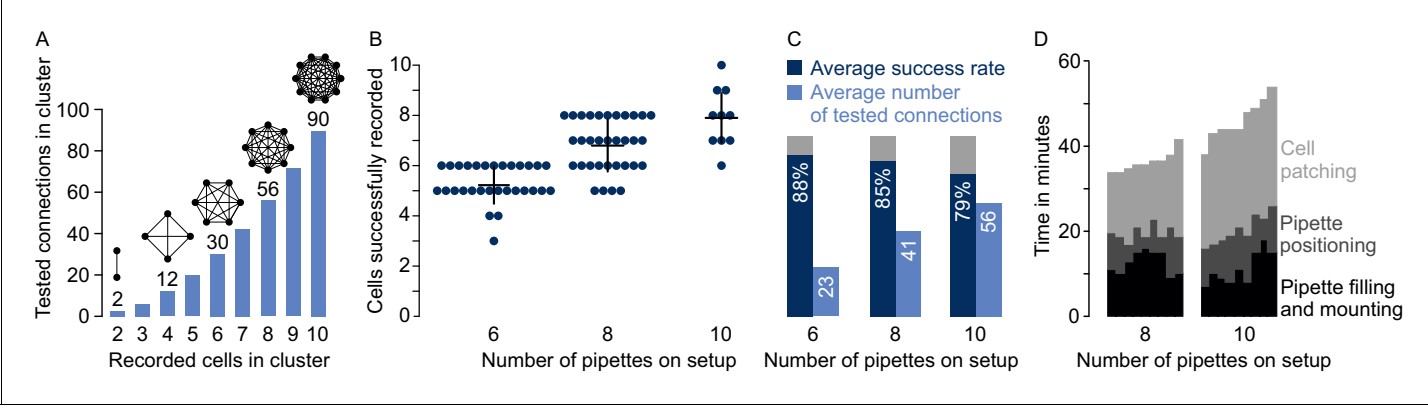

**Figure 4.** Performances on multipatch setups. (**A**) Bar graphs depicting maximum number of testable connections for increasing number of simultaneously recorded cells. (**B**) Dot plot of number of simultaneously recorded cells from rodent brain slice experiments using different multipatch setups. Black crosses indicate mean and standard deviation. (**C**) Performance parameters derived from B. Average success rate represents the average size of recorded clusters in relation to the maximally possible cell number given the available pipettes on each setup. Average numbers of tested connections are derived from the number of tested connections from each experiment. Note the decrease in success rate and slowing increase of tested connections when the number of pipettes is increased. (**D**) Stacked bar graph of time needed for individual preparatory steps of single experiments for the eight-manipulator setup (n = 9) and the 10-manipulator setup (n = 9). Source data provided in *Figure 4—source data 1* and *Figure 5—source data 1*.

The online version of this article includes the following source data for figure 4:

**Source data 1.** Source Data - experimental times.

implement this for n channels would require $2^n$ different stimulation protocols and restarting the recording for every newly patched neuron would also create multiple dissimilar recording files. However, these issues could be addressed through automated generation of recording protocols and data analysis. In our connectivity experiments, after patching all cells, we switch all channels to current clamp mode, manually adjust the bridge balance and capacitance compensation and then record the electrophysiological parameters including firing patterns. This initial assessment takes up to 5 min. Afterwards, we perform a connectivity screening protocol which records up to 50 sweeps in 10 to 12.5 min depending on the number of neurons in the cluster.

## Pipette cleaning strategies for multipatch experiments

The automated pipette cleaning system can be applied in different ways to improve multipatch experiments. As demonstrated above, multipatch setups will often fail to achieve their full potential due to a decreasing success rate with increasing pipette numbers. The automated pressure and cleaning system can solve this problem and increase the yield through more complete cluster recordings. Pipettes from failed recording attempts can be subject to immediate cleaning and reuse, even simultaneously to ongoing patch attempts with the next pipette (*Figure 5A*). This clean-to-complete strategy increased the average cluster size on our setups with 8 and 10 pipettes to 7.8 ± 0.4 (97% ± 5% success rate, n = 16) and 9.2 ± 0.6 (92% ± 6% success rate, n = 9), respectively (*Figure 5B*). After recording of a full cluster, the cleaning system can be used to probe synaptic connections to cells outside of the initial cluster (clean-to-extend, *Figure 5D*). Multiple pipettes can be selected and cleaned to establish new patch-clamp recordings with other cells of interest (*Figure 5E*). This clean-to-extend approach allows screening of many additional synaptic connections with minimal time investment. The number of additionally probed connections ($c_{new}$) is dependent on the number of newly patched ($n_{new}$) and maintained neurons ($n_{old}$): $c_{new} = 2 \times (n_{new} \times n_{old}) + n_{new}(n_{new} - 1)$. Using both strategies during rodent experiments on the eight-pipette setup with two to recorded slices per animal, we could increase the number of tested connections from 140 ± 24 to 244 ± 52 on average per animal (*Figure 5D*, mean ± standard deviation, n = 6 animals). To assess possible bias through the locations of repatched cells, we analyzed the distances of 1324 probed connections (662 pairs) from these experiments and compared them across different rounds of recording after performing up to two clean-to-extend sessions (*Figure 5—figure supplement 1*). The distance distribution in the initial cluster (median 84 μm, IQR 61–110 μm,

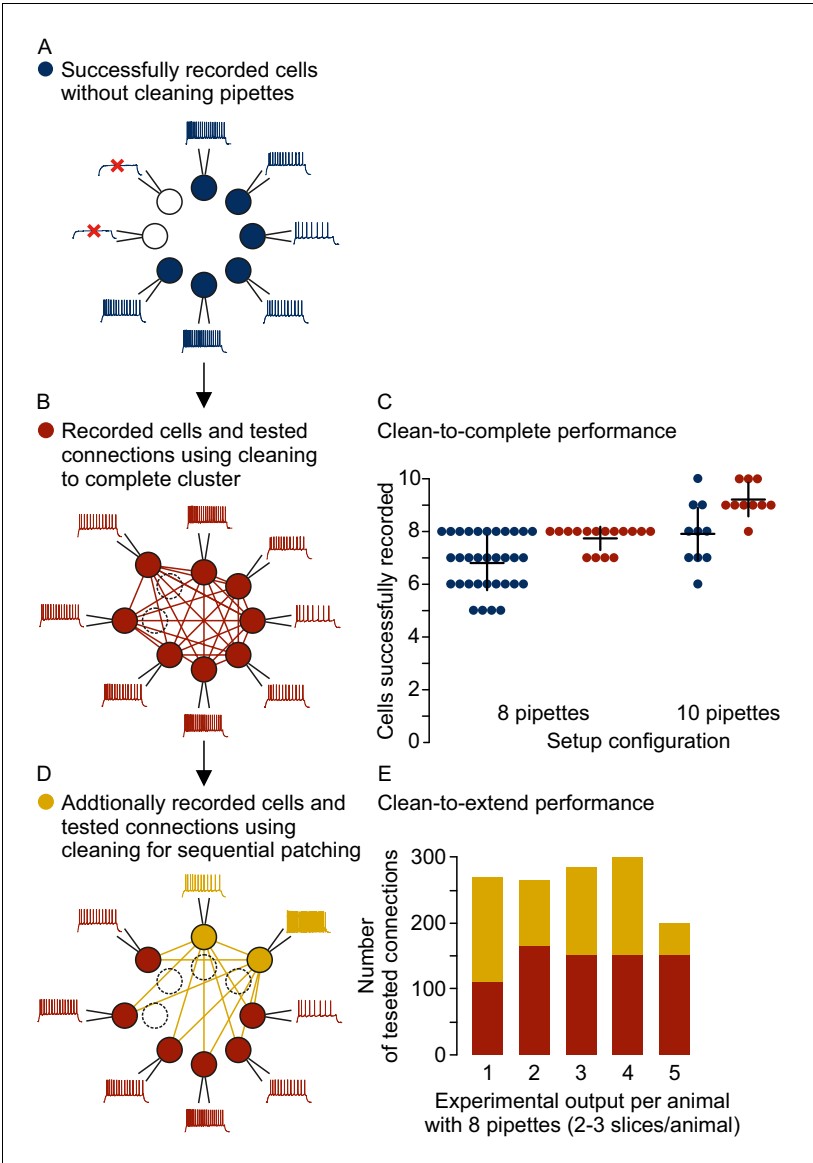

**Figure 5.** Different strategies for cleaning pipettes. (**A**) Scheme of successful whole-cell recordings on six pipettes (blue circles with action potentials) and failed patch attempts on two pipettes (white circles without action potentials). (**B**) Clean-to-complete: The two failed pipettes were cleaned and successful recordings were established from two neighbouring cells. Red circles represent a cluster that has been subject to pipette cleaning after failed patch attempts and the red lines indicate the tested synaptic connections. (**C**) Dot plot comparing the number of successfully recorded neurons in clusters without cleaning (blue) and with cleaning (red). Black crosses indicate mean and standard deviation. See *Figure 5—source data 1*. (**D**) Clean-to-extend: After recording of a cluster, individual pipettes can be cleaned and used to patch additional cells (yellow circles). Yellow lines indicate the synaptic connections that could be tested due to this approach. (**E**) Stacked bar graph depicting the number of tested connections of the initially recorded cluster with clean-to-complete (red) and the number of additionally tested connections after clean-to-extend (yellow) from five animals. Data were extracted from experiments in the rat presubiculum on the 8-manipulator setup. Two to three slices were analyzed from each animal in a time window of 5 to 6 hr. Intersomatic distance distribution of these experiments are shown in the figure supplement to this figure. Source data provided in *Figure 5—source datas 1* and *2*.

The online version of this article includes the following source data and figure supplement(s) for figure 5:

**Source data 1.** Clean2complete success rate.
**Source data 2.** Clean2extend connections.
**Figure supplement 1.** Intersomatic distance distribution Box plots depicting the intersomatic distances of tested connections from rat presubiculum experiments in different clean2extend recording rounds.

**Table 1.** Hardware components of 10-manipulator setup.

| Hardware | Manufacturer | Model |
|---|---|---|
| five dual patch-clamp amplifiers | Molecular Devices | MultiClamp 700B<br>CV-7B headstage |
| Data acquisition system | Cambridge Electronic Design | Power1401-3A<br>+ Signal Expansion (2701–5) |
| Stimulation routing device | Custom-made | Design in supplement |
| 10 Micromanipulators | Sensapex | u-Mp Micromanipulator |
| Microscope | Nikon | Eclipse FN1 |
| Motorized stage for microscope | Scientifica | UMS-2550P for x-/y-axis<br>Stepper motor for z-axis drive |
| Camera | Hamamatsu | Orca-Flash 4.0 |
| LED infrared light source | Thorlabs | M780L3 |
| four height adjustable poles | Thorlabs | P250/M, PB1, C1515/M |
| Active vibration isolation table | Accurion | Halcyonics_i4large |
| Peristaltic pump | Gilson | Minipuls 3 |
| Thermostat | Multichannel systems | TC01 |
| Stage for manipulators | Custom-made | Design in supplement |
| Recording chamber | Custom-made | Design in supplement |

278 pairs) was similar and not systematically shifted compared to the second (median 85 µm, IQR 62–109 µm, 218 pairs) and third round of recording (median 84 µm, IQR 60–107 µm, 166 pairs). Clean-to-extend can also be used to selectively patch more cells of interest while a set of cells are maintained. This would enable exploration of specific connectivity patterns and a more elaborate degree analysis of specific cells.

Combining multipatch setups with the optimizations and cleaning strategies allows for extensive and efficient analysis of human microcircuits. Due to the scarce availability of human tissue from epilepsy resection surgery, a highly efficient usage of this material is desirable. With two rounds of clean-to-extend, we were able to record electrophysiological properties and the synaptic connections of up to 17 neighboring neurons in one human slice (*Figure 6A*). In this experiment, we maintained two neurons (1.1 and 1.2) across all further patch attempts providing information about their out- and in-connections to the other 16 neurons, respectively. This approach can unveil complex connectivity patterns between local neurons exceeding simple pairwise statistics (*Figure 6B–D*). Using two setups simultaneously, we were able to record from up to 99 neurons and thereby probe 300 to 700 potential synaptic connections from individual patients within the first 24 hr after slicing. These sample sizes acquired from single patients are comparable to datasets on connectivity within single cortical layers that were generated by pooling several animals or patients (*Lefort et al., 2009*; *Jiang et al., 2015*; *Seeman et al., 2018*). We found excitatory connectivity between pyramidal cells in human cortical layer 2/3 ranging from 12.9% to 17.7% which is similar to those reported previously (*Boldog et al., 2018*; *Seeman et al., 2018*; *Szegedi et al., 2016*). The inter-individual differences were not statistically different from each other (Pat. 1 vs 2, p=0.08; Pat. 1 vs 3, p=0.26; Pat. 2 vs 3, p=0.41) while they were statistically equivalent within a range of −5 and 9.5% (Pat. 1 vs 2, 4.9% [0.3% 9.5%], Pat. 1 vs 3, 3.1% [-1.5% 7.7%], Pat. 2 vs 3, −1.8% [−5% 1.5%], proportion difference [90% confidence interval of difference]). This substantial improvement in multipatch experiment yield has the potential to facilitate systematic analysis of complex network properties at the level of individuals in humans and other species.

## Discussion

In this report, we demonstrate that equipping an in vitro multipatch setup with an automated pipette cleaning system and adding further optimizations increases the experimental yield and generates sufficient data for microcircuit analysis within single patients. This represents a substantial

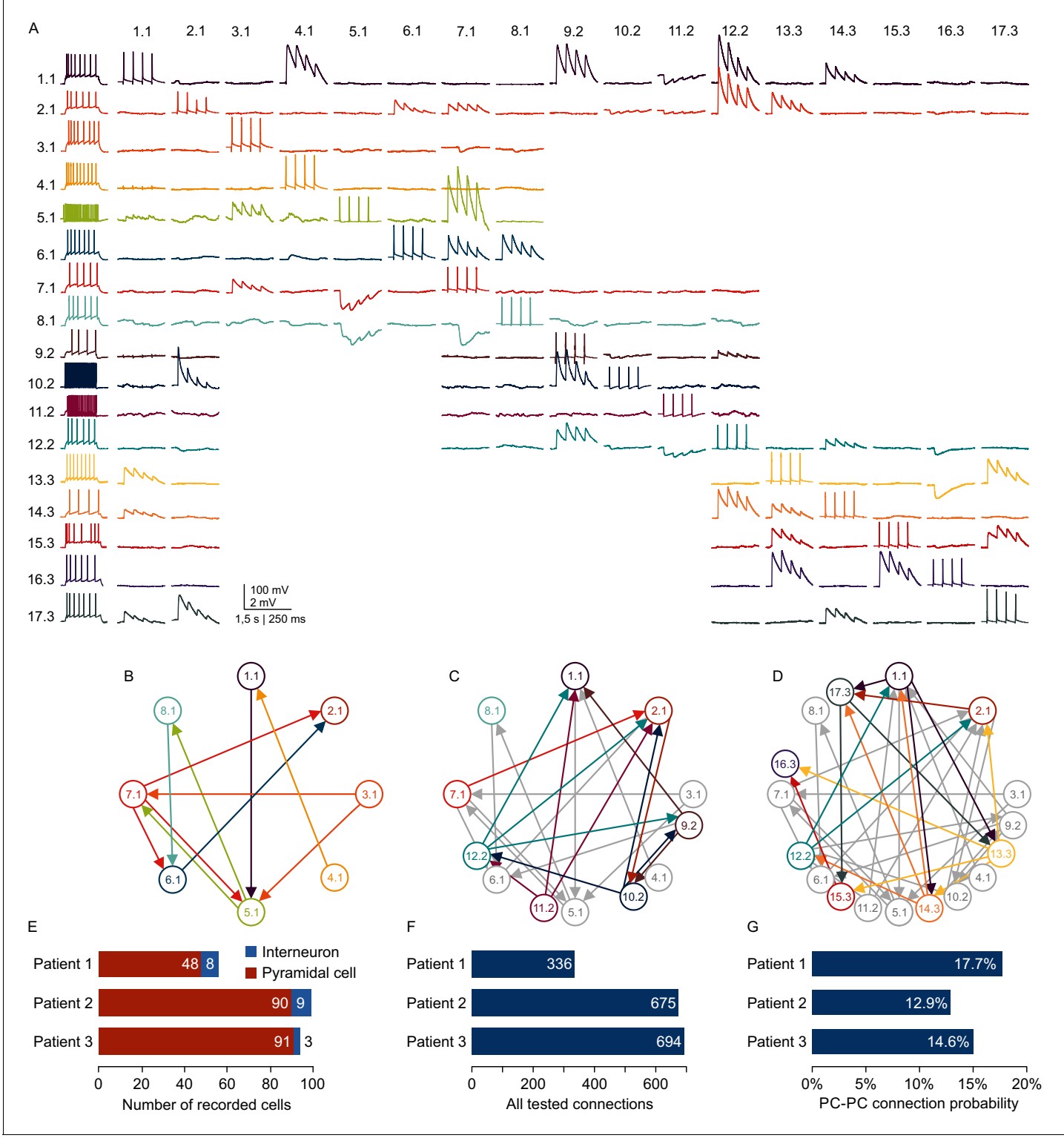

**Figure 6.** Microcircuit analysis in human slices. (**A**) Matrix of averaged voltage traces from 17 neurons in one acute human slice recorded on the -manipulator setup with two rounds of clean-to-extend. Left column shows the firing pattern of the recorded neurons. In the first session eight neurons were patched simultaneously (cells numbered 1.1–8.1). Traces recorded from one cell are shown in a row with the same color. Four action potentials were elicited in each neuron consecutively (diagonal of the matrix). The postsynaptic responses of the other neurons are aligned in the same column. After recording of the first full cluster of 8eight neurons, four pipettes were cleaned and additional neurons in the vicinity were patched and recorded with the same stimulation protocol (9.2–12.2). After the second recording session, another five pipettes were subject to cleaning and five new neurons

*Figure 6 continued on next page*

*Figure 6 continued*

were patched, while the pipettes on neuron 1.1, 2.1 and 12.2 were not removed. This allowed screening of additional connections among the neurons from the third recording session (13.3–17.3) and also connections between neurons from previous recording sessions (1.1, 2.1 and 12.2). Scale bar: Horizontal 1.5 s for firing pattern, 250 ms for connection screening. Vertical 100 mV for action potentials, 2 mV for postsynaptic traces. (B) Connectivity scheme of all neurons from the first recording session with arrows indicating a detected synaptic connection. (C) Scheme of connections after first cleaning round. Colored arrows and circles indicate neurons and connections recorded in the second session. Neurons and connections of the first recording session are shown in gray. (D) Scheme of all recorded neurons and detected connections after two cleaning rounds. Neurons and connections recorded in this third session are colored. Neurons and connections from previous recording sessions are shown in gray. In this slice, a total of 38 synaptic connections were detected out of 150 tested connections. (E) Bar graphs depicting number of recorded interneurons and pyramidal cells from three patients recorded within the first 24 hr after slicing. (F) Number of tested connections in each patient recorded within the first 24 hr. (G) Connection probability between pyramidal cells calculated by the number of found to tested connections recorded within the first 24 hr. See *Figure 6—source data 1*.

The online version of this article includes the following source data for figure 6:

**Source data 1.** Human multipatch.

increase in performance and experimental output compared to previous studies on synaptic connectivity, which usually require pooling of data across multiple subjects to achieve statistically significant results (*Seeman et al., 2018*; *Thomson and Lamy, 2007*). With 300 to 600 probed connections per patient, variant and invariant parameters can be identified and motivate studies that correlate microcircuit properties to individual patient characteristics as has been shown for dendritic morphology and action potential kinetics (*Goriounova et al., 2018*). Applied to animal studies, this approach can reduce the number of animals needed to reach statistically significant findings. This is especially important when investigating animal models where ethical considerations prohibit large cohorts due to harmful interventions.

The possibility to further explore the connectivity of neurons through clean-to-extend allows microcircuit analysis on a larger scale. While manual replacement of pipettes is possible, the risk of mechanical disruption of the recordings up to now prevented extensive practice of patching cells sequentially. Therefore, studies on synaptic connectivity have been limited by the number of available manipulators. Increasing the sample size of recorded neurons and tested synaptic connections through clean-to-extend now enables a more comprehensive analysis of the microcircuit. This is especially relevant as theoretical work has identified higher order network parameters such as triadic motifs and simplices as relevant topological constraints for microcircuit computation (*Perin et al., 2011*; *Reimann et al., 2017*; *Song et al., 2005*). A minimum of to four neurons per cluster is necessary for triplet analyses, while increasing the number leads to larger sample sizes and enables higher dimensional parameters. This increase in sample size per cluster also enables a better assessment of slice to slice variability, which can help to control for methodological artefacts or confounding biological parameters. However, our impression is that increasing setups beyond 10 manipulators would come with a serious trade-off regarding experimental time and success rate. Clean-to-extend enables increasing the number of probed connections of neurons beyond this limitation which is important for network statistics such as the sample degree correlation and can be utilized for further novel graph statistics (*Vegué et al., 2017*).

Underestimation of synaptic connectivity in slice experiments is an important concern and subject to ongoing debate (*Jiang et al., 2015*; *Jiang et al., 2016*; *Barth et al., 2016*). Axons and dendrites of neurons close to the slice surface can be severed in the slicing process, potentially leading to false negatives. Anatomical reconstructions can help to assess the extent of axon cutting and refine the optimal slicing angle while simulations of slicing effects on connection probability have shown a correction factor between 1.3 to 1.7 (*Levy and Reyes, 2012*; *Jiang et al., 2016*). On the other hand, in vivo recordings with no cutting effects found a connection probability of 6.7% between pyramidal cells in the mouse barrel cortex compared to 9.3% in slice experiments (*Jouhanneau et al., 2018*; *Lefort et al., 2009*). Another potential limitation stems from missing very weak synaptic potentials when the signal-to-noise ratio is low. To address this, a previous study has used simulations of background noise to determine the detection limits, reporting a correction factor of 1.1 for connection probability (*Seeman et al., 2018*). To generate valid estimates of connection probabilities in slice recordings, it is therefore important to avoid superficial cells near the slice surface, control for intersomatic distances and ensure good recording quality.

We addressed technical challenges in establishing and operating a highly advanced multipatch setup and provide extensive documentation on how to implement optimizations making this method easier to handle. Patch-clamp electrophysiology is a widespread and established method, but operating multiple manipulators still requires skilled researchers with long training. The low-cost and user-friendly solutions presented here can be used to upgrade existing patch-clamp setups with single or multiple manipulators. This could enable other labs to adopt the multi-neuron approach and use the optimizations to reduce experimental errors and generate results faster. Additionally, semi-automated pipette positioning as well as repetitive pipette cleaning without the need for manual replacement can help novel experimentalists to focus on the crucial steps of cell patching and may reduce the training time.

While fully automated approaches have been introduced for in vitro and in vivo patch-clamp experiments (*Annecchino et al., 2017*; *Desai et al., 2015*; *Kodandaramaiah et al., 2018*; *Kodandaramaiah et al., 2012*; *Kolb et al., 2019*; *Suk et al., 2017*), the advantage for fully automated multipatch is less clear. We found that manual adjustments during the visually guided approach of the cell and the fine-tuned application of pressure to obtain a whole-cell configuration yielded higher success rates (88%/85 % / 79% on the 6-/8-/10-manipulator setup) compared to a fully automated system with pipette cleaning and machine vision which reported a success rate of 43% to 51% (*Wu et al., 2016*; *Kolb et al., 2019*). Our semi-automated pipette positioning approach was faster (53/55 s per pipette on 8-/10-manipulator setup) than the fully automated method described in *Wu et al. (2016)* (103 s using one pipette) while the time needed to manually approach the cell and establish a whole-cell configuration was slightly higher (126/154 s per pipette on 8-/10-manipulator setup vs 120 s). Further optimized algorithms with better performance could be combined with pipette cleaning to improve the success rate and speed of multipatch experiments. However, we currently believe that fully automated multipatch experiments would require a disproportionate amount of effort for the resulting increase in performance. Also, slice exchange and cell selection would still require human presence.

By overcoming the presented technical limitations, other aspects of the experimental pipeline become a bottleneck and need to be addressed. Automated data analysis is an essential step to leverage the increased experimental throughput. However, as this report highlights technical developments that should be applicable to other groups, we refrained from detailing our data analysis which is still an ongoing effort. For connectivity studies, a previous study has demonstrated an automatic synapse detection algorithm using a support vector machine classifier that achieved an accuracy of 95% (*Seeman et al., 2018*). Alternatively, semi-automated commercial solutions such as the batch analysis function of pClamp 11 (Molecular Devices) or custom Signal scripts (Cambridge Electronic Design) can be used but necessitate manual curation. After the primary trace analysis, adjacency matrices or lists representing the synaptic connections in each cluster can be generated. Further properties, such as synaptic strength or latency of these connections, can also be represented in adjacency matrices or lists. Another list containing the cell identities and properties (anatomical position and physiological parameters) should be generated for each cluster. Such data formats can be used for further graph analysis and represent a suitable intermediate format for data sharing on databases, such as Neurodata without borders (*Teeters et al., 2015*). Furthermore, several steps during the experiment need manual setting through the MultiClamp 700B Commander (Molecular Devices), such as switching between voltage and current clamp mode or compensation of bridge balance and pipette capacitance. To allow fully automated setting of the amplifiers and running of specific stimulation protocols, custom-written programs need to be able to control the amplifiers and the data acquisition board. We believe that using commercially available software allows greater experimental flexibility with minimal time investment while further automation would be worth considering for standardized high-throughput experiments.

Compared to other methods, the multipatch approach provides superior access to cellular and synaptic properties with a trade-off regarding the size of the sampled network. All-optical approaches using channelrhodopsins to stimulate individual neurons and fluorescent calcium or voltage indicators to measure their responses can be used to probe connectivity of a higher number of cells than the multipatch approach (*Emiliani et al., 2015*). Simultaneous optical stimulation and recording using calcium indicators have been performed in up to 20 neurons (*Packer et al., 2015*). However, calcium transients cannot resolve single action potentials or subthreshold postsynaptic signals, potentially missing monosynaptic connections. Voltage indicators can resolve subthreshold

signals with higher temporal resolution than calcium imaging and have been shown to detect mono-synaptic connections combined with optical stimulation in organotypical slices (*Hochbaum et al., 2014*). But technical challenges such as imaging speed and high illumination intensities still limit its application for large network investigations in acute brain slices. While calcium imaging can also be applied in vivo, multipatch in this condition is technically very challenging and thus far limited to a maximum of four pipettes (*Jouhanneau et al., 2018*; *Kodandaramaiah et al., 2018*). Patch-clamp recordings can furthermore be combined with presynaptic stimulation using extracellular electrodes, channelrhodopsin or glutamate uncaging to further map synaptic inputs from more neurons. Stimulation through a high-density multielectrode array or two-photon glutamate uncaging allows mapping of monosynaptic inputs from 10 to 50 presynaptic neurons onto a single patched neuron (*Fino and Yuste, 2011*; *Jäckel et al., 2017*), while two-photon optogenetic stimulation can probe connections from up to 200 presynaptic neurons in a single experiment (*Izquierdo-Serra et al., 2018*). These approaches do not resolve electrophysiological and anatomical properties and further connectivity of the presynaptic neurons, but they can be used to complement the multipatch approach. Further-more, other fundamental properties of synaptic transmission such as synaptic strength and plasticity are only accessible through postsynaptic patch-clamp recording and presynaptic electrical or opto-genetic stimulation. Finally, multipatch experiment do not need genetically modified cells and can be applied on brain slices from a variety of species.

Taken together, the experimental advances presented here enables highly efficient extraction of microcircuit parameters from human cortical tissue even at the level of individual patients. The potential to analyzse reciprocal connections of more than 10 neurons in one cluster will help to develop more sophisticated subgraph metrics, strengthening the inference onto the underlying microcircuit. Finally, this versatile method can be applied to various species to uncover overarching principles of microcircuit topology.

## Materials and methods

### Human and animal tissue

Human tissue was acquired from temporal lobe resections of patients with pharmacoresistant tem-poral lobe epilepsy. All patients gave a written consent for the scientific use of the resected tissue. All procedures adhered to ethical requirements and were in accordance to the respective ethical approval (EA2/111/14).

Performance data of rodent multipatch experiments without cleaning were extracted from previ-ous studies on mouse subiculum (six-pipette setup; *Böhm et al., 2018*) and rat presubiculum (eight-pipette setup; *Peng et al., 2017*). Performance data with cleaning on the eight-manipulator setup were extracted from unpublished rat presubiculum experiments and all performance data on the 10-manipulator setup were extracted from unpublished rat motor cortex experiments. Experimental times were documented during a subset of experiments. Acute brain slices were prepared from 19 to 35 days old transgenic Wistar rats expressing Venus-YFP under the VGAT promotor (*Peng et al., 2017*; *Uematsu et al., 2008*) or from 21 to 42 days old male C57BL/6N mice (*Böhm et al., 2015*). Animal handling and all procedures were carried out in accordance with guidelines of local authori-ties (Berlin, [T0215/11], [T0109/10]), the German Animal Welfare Act and the European Council Directive 86/609/EEC.

### Slice preparation

Experimental procedure on human tissue was as previously described (*Lehnhoff et al., 2019*). In brief, tissue samples were collected at the operating theatre and transferred to the laboratory within 30 to 40 min in cooled sucrose based aCSF enriched with carbogen (95% O2, 5% CO$_2$). They were cut in ice-cold sucrose aCSF containing (in mM): 87 NaCl, 2.5 KCl, 3 MgCl$_2$, 0.5 CaCl$_2$, 10 glucose, 75 sucrose, 1.25 NaH$_2$PO$_4$, and 25 NaHCO$_3$ (310 mOsm), enriched with carbogen (95% O2, 5% CO$_2$). After removal of residual pia, the tissue was cut into 300 to 400-µm-thick slices and subse-quently stored in sucrose aCSF solution heated to 34°C for 30 min recovery. Slice thickness was 400 µm. In a subset of experiments, an antibiotic was added to the incubation solution (minocycline 2 nM). Subsequently and until recording, the slices were stored at room temperature. Whole-cell recordings were performed at 34°C under submerged conditions, the bath chamber was perfused

with an aCSF solution containing (in mM): 125 NaCl, 2.5 KCl, 1 $MgCl_2$, 2 $CaCl_2$, 10 glucose, 1.25 $NaH_2PO_4$, and 25 $NaHCO_3$ (300 mOsm). Patch pipettes were pulled from borosilicate glass capillaries (2 mm outer/1 mm inner diameter; Hilgenberg) on a horizontal puller (P-97, Sutter Instrument Company) and filled with intracellular solution containing (in mM): 130 K-gluconate, 2 $MgCl_2$, 0.2 EGTA, 10 $Na_2$-phosphocreatine, 2 $Na_2ATP$, 0.5 $Na_2GTP$, 10 HEPES buffer and 0.1% biocytin (290–295 mOsm, pH adjusted to 7.2 with KOH).

The rat experiments were performed as previously reported (*Peng et al., 2017*). After isoflurane anaesthesia, decapitation and removing parts of the skin and skull, the brain was submerged in an ice-cold sucrose artificial cerebrospinal fluid solution containing (in mM): 80 NaCl, 2.5 KCl, 3 $MgCl_2$, 0.5 $CaCl_2$, 25 glucose, 85 sucrose, 1.25 $NaH_2PO_4$ and 25 $NaHCO_3$(320–330 mOsm), enriched with carbogen (95% $O_2$, 5% $CO_2$). Brain slices of 300 µm thickness were cut on a Leica VT1200 vibratome (Leica Biosystems) and subsequently stored in the sucrose aCSF solution. After a recovery period of 30 min at 30˚C, the slices were stored at room temperature. Whole-cell recordings were performed under submerged conditions, the bath chamber was perfused with an aCSF solution containing (in mM): 125 NaCl, 2.5 KCl, 1 $MgCl_2$, 2 $CaCl_2$, 25 glucose, 1.25 $NaH_2PO_4$, and 25 $NaHCO_3$ (310–320mOsm). The solution was enriched with carbogen (95% $O_2$, 5% $CO_2$) and heated to 34˚C. Patch pipettes were pulled in the same way as for the human slice experiments and filled with intracellular solution containing (in mM): 130 K-gluconate, 6 KCl, 2 $MgCl_2$, 0.2 EGTA, 5 $Na_2$-phosphocreatine, 2 $Na_2ATP$, 0.5 $Na_2GTP$, 10 HEPES buffer and 0.1% biocytin (290–300 mOsm, pH adjusted to 7.2 with KOH).

## Recording protocol and data analysis

Hardware components used for visualization and electrophysiological recordings are listed in *Table 1*. Data were low-pass filtered at 6 kHz using the built in four-pole Bessel filter and digitized at a sampling rate of 20 kHz. For synaptic connectivity screening, all cells were recorded in current clamp mode and held near –60 mV by means of constant and adjusting current injection. Four action potentials were elicited in each cell at 20 Hz with 1–4 nA for 1–3 ms and postsynaptic responses in the other cells were detected in the averaged traces from 30 to 50 sweeps. The current amplitude and duration needed to elicit a single action potential was determined in a separate stimulation adjustment protocol: We applied increasing current from 0.5 to 4 nA at 0.5 nA steps. Stimulation duration was increased when no action potential could be elicited at 1 ms. The second lowest amplitude at the specific duration that could elicit an action potential was chosen and manually entered in the connectivity screening protocol. Successful recordings were defined by a sufficiently high seal-resistance, resting membrane potentials of the cells more negative than −50 mV (not corrected for liquid junction potential), long-term recording stability, and the ability of the neurons to produce characteristic action potential patterns during sustained step depolarisations.

Control experiments in rodent tissue for comparing electrophysiological properties before and after pipette cleaning were performed in layer 2/3 motor cortex of 21- and 22-day-old Wistar rats. We recorded 81 neurons in 12 brain slices using 28 pipettes with up to two sequential cleaning rounds and analyzed the cellular and synaptic physiology of these cells. To guarantee a homogeneous sample of comparable and healthy pyramidal neurons, we excluded interneurons and nine pyramidal neurons with a resting membrane potential more positive than −60 mV. These nine depolarised cells were detected with equal probability on fresh and cleaned pipettes (3/28 cells with fresh pipettes, 2/28 cells after first cleaning and 4/28 cells after second cleaning). One recording after the second cleaning was discarded due to high access resistance (129 MΩ), in all other 80 patch attempts, we were able to establish a giga-ohm seal resistance and acquire a stable whole-cell configuration with an access resistance below 40 MΩ. Thus, the success rate of patch attempts after cleaning in these experiments was 98% (53 successful whole-cell recordings out of 54 patch attempts).

Control experiments in human tissue were acquired from two epilepsy patients undergoing frontal and temporal cortical resection. For the comparison between different cells recorded with fresh and cleaned pipettes, 33 cells were included from five slices in which both conditions were measured. For the comparison between cells recorded with fresh pipettes and repatched with cleaned pipettes we recorded 15 cells in five slices. To guarantee a homogenous sample of comparable and healthy neurons, we excluded six neurons with a resting membrane potential more positive than −60 mV after repatch. We believe that this deterioration is more likely due to the repatching

procedure than the pipette cleaning. Comparison between five cells recorded with fresh pipettes and repatched with freshly pulled new pipettes were acquired from one slice. Comparison of properties of the same neuron recorded before and after rinsing in ACSF were acquired from 22 neurons from four slices. Here, we patched clusters of five to six neurons with fresh pipettes and recorded their intrinsic properties and the synaptic connections between them. Then, we simulated a typical experiment by subjecting the pipettes with no cells patched to the cleaning procedure. Afterwards, we recorded the intrinsic properties and the synaptic connections again between the cells that were kept on the respective pipettes. Here, we excluded three neurons with a resting membrane potential more positive than −60 mV that were patched in the fresh ACSF to ensure a sample of comparable and healthy neurons.

Experiments were performed on the eight-manipulator setup with Scientifica PatchStar manipulators, Digidata 1550 digitizer and pClamp 10 acquisition software (Molecular Devices) and on the 10-manipulator setup with Sensapex manipulators, Power1401-3A digitizer and the Signal acquisition software (Cambridge Electronic Design). All trace analysis for the cellular and resistance parameters were performed using custom written scripts in the Signal 6 software (Cambridge Electronic Design). Signal scripts can be found at https://github.com/neurocharite/multipatch. (*Peng and Mittermaier, 2019*; copy archived at https://github.com/elifesciences-publications/multipatch). Access and input resistance were measured in voltage clamp configuration using the peak and steady state value elicited by a 200 ms long 10 mV pulse. The resting membrane potential was determined using a 100 ms baseline. Action potential parameters were analyzed based on the second action potential at the rheobase. The voltage value at the point where the trace exceeds a slope of 10 mV/ms relative to the baseline was determined as the action potential threshold. The action potential amplitude was calculated as the difference between the action potential peak and the threshold. Spontaneous EPSPs were analysed in a 2 s current clamp trace at resting membrane potential using the Stimfit software (Version 0.15.8, *Guzman et al., 2014*). To detect spontaneous EPSPs, we performed template matching based on a template scaling method (*Clements and Bekkers, 1997*) with subsequent manual curation of false-positive and false-negative events. For each cell, the median EPSP amplitude and mean EPSP frequency was determined.

## Statistical analysis

To assess the changes in physiological properties before and after pipette cleaning, we calculated the mean relative difference and the 90% confidence interval of it to determine the level of statistically significant equivalence. Statistical difference for independent samples was assessed using the Mann-Whitney U test while comparison of dependent samples was analyzed with the Wilcoxon signed-rank test. Changes in properties across three conditions (rat slices: fresh pipette, 1x cleaning, 2x cleaning) were statistically tested using a repeated measures ANOVA. Data processing and analysis was performed using MATLAB R2018a (MathWorks) and an online available code allowing the two one-sided test (TOST) for equivalence (*Rogers et al., 1993*). Statistical comparison between connection probabilities were analyzed using the Fisher's exact test. The confidence interval for the difference as the range of statistically significant equivalence was calculated through a simple asymptotic method without continuity correction ('Wald method', *Newcombe, 1998*):

$$confidence\ interval\ for\ difference = (p_1 - p_2) \pm z\sqrt{\frac{p_1(1-p_1)}{n_1} + \frac{p_2(1-p_2)}{n_2}}$$

## Acknowledgements

We are grateful to the patients for providing the tissue and thank P Fidzinski and M Holtkamp for clinical organization and assistance. We also thank L Faraj for support during the creation of the pressure system assembly guide and I Vida for providing the transgenic rats. We thank the mechanical workshop of the Charité-Universitätsmedizin Berlin for technical assistance and fabrication of the custom-made components. We thank Jochen Winterer and Rosanna Sammons for helpful comments on an earlier version of the manuscript.

## Additional information

### Funding

| Funder | Grant reference number | Author |
|---|---|---|
| Deutsche Forschungsge-meinschaft | EXC 2049: Project number 390688087 | Jörg Rolf Paul Geiger |

The funders had no role in study design, data collection and interpretation, or the decision to submit the work for publication.

### Author contributions

Yangfan Peng, Conceptualization, Resources, Data curation, Software, Formal analysis, Validation, Investigation, Visualization, Methodology, Writing—original draft, Project administration, Writing—review and editing; Franz Xaver Mittermaier, Conceptualization, Resources, Data curation, Software, Formal analysis, Validation, Investigation, Visualization, Methodology, Writing—original draft, Writing—review and editing; Henrike Planert, Data curation, Investigation, Writing—review and editing; Ulf Christoph Schneider, Resources, Project administration; Henrik Alle, Conceptualization, Resources, Software, Methodology, Writing—original draft, Writing—review and editing; Jörg Rolf Paul Geiger, Conceptualization, Supervision, Funding acquisition, Writing—original draft, Project administration, Writing—review and editing

### Author ORCIDs

Yangfan Peng (iD) https://orcid.org/0000-0002-0317-1353
Franz Xaver Mittermaier (iD) https://orcid.org/0000-0003-2258-3051
Jörg Rolf Paul Geiger (iD) https://orcid.org/0000-0001-9552-4322

### Ethics

Human subjects: All patients gave a written consent for the scientific use of the resected tissue. All procedures adhered to ethical requirements and were in accordance to the approval of the ethics committee of the Charité - Universitätsmedizin Berlin (EA2/111/14).
Animal experimentation: Animal handling and all procedures were carried out in accordance with guidelines of local authorities (Berlin, [T0215/11], [T0109/10]), the German Animal Welfare Act and the European Council Directive 86/609/EEC.

### Decision letter and Author response

Decision letter https://doi.org/10.7554/eLife.48178.sa1
Author response https://doi.org/10.7554/eLife.48178.sa2

## Additional files

### Supplementary files

• Transparent reporting form

### Data availability

All data generated or analysed during this study are included in the manuscript and supporting files. Source data files have been provided for Figures 3, 4, 5 and 6.

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

## Appendix 1

### Pressure and cleaning system

### Overview of components

**Appendix 1—table 1.** Table of electronic parts referring to *Appendix 1—figure 1* including part number and cost.

| # | Electronic parts | Amount/pieces | Supplier/producer* | Part number* | Unit cost €* | Total cost €* |
|---|---|---|---|---|---|---|
| a | 4-channel relay board 5 V | 1 | sertronics | RELM-4 | 3.19 | 3.19 |
| b | 16-channel relay board 12 V | 2 | sertronics | RELM-16 | 16.72 | 33.45 |
| c | Arduino Mega 2560 microcontroller | 1 | reichelt | ARDUINO MEGA | 26.81 | 26.81 |
| d | 20 position male to female jumper wire, 20 cm length | 2 | reichelt | DEBO KABELSET | 3.24 | 6.47 |
|   | power supply 80W, 12 V / 6,67 A | 1 | reichelt | MW GST90A12 | 25.59 | 25.59 |
| e | aluminium angle 20 × 20×2 mm | 20 cm | alusteck | W20202 | 1.58 | 1.58 |
| f | ON/OFF switch, 250 V / 6 A | 1 | reichelt | MAR 1821.1101 | 1.67 | 1.67 |
| g | DC barrel power connector, 2.5 mm center pole diameter | 1 | reichelt | HEBL 25 | 0.34 | 0.34 |
| h | metal spacer hexagonal, male/female M3, 10 mm | 14 | reichelt | DA 10 MM | 0.11 | 1.53 |
| i | metal spacer hexagonal, male/female M3, 25 mm | 24 | reichelt | DA 25 MM | 0.17 | 4.03 |
| j | plastic spacer, M3, 10 mm | 4 | reichelt | DK 10 MM | 0.03 | 0.12 |
| j | plastic spacer, M3, 20 mm | 2 | reichelt | DK 20 MM | 0.03 | 0.06 |
| k | screw terminal block, PCB mount, 12 pole | 6 | reichelt | RND 205–00242 | 0.83 | 4.99 |
| l | stripboard, pitch corresponding to screw terminal | 2 | reichelt | H5SR160 | 1.35 | 2.69 |
|   | acrylic plate, 31 × 27 cm, 10 mm thickness | 1 | Express-zuschnitt | AGS-100-TRA | 11.20 | 11.20 |
|   | acrylic plate, 24 × 19 cm, 5 mm thickness | 1 | Express-zuschnitt | AGS-50-TRA | 5.97 | 5.97 |
|   | hexagonal nut, M3 | 100 | reichelt | SK M3 | 0.83 | 0.83 |
|   | hexagonal nut, M2 | 100 | reichelt | SK M2 | 1.81 | 1.81 |
|   | bolt, M2, 20 mm | 100 | reichelt | SZK M2 × 20 | 7.90 | 7.90 |
|   | bolt, M3, 10 mm | 100 | reichelt | SZK M3 × 10 | 1.60 | 1.60 |
|   | bolt, M3, 25 mm | 100 | reichelt | SZK M3 × 25 | 2.27 | 2.27 |
|   | bolt, M3, 35 mm | 100 | reichelt | SZK M3 × 35 | 3.78 | 3.78 |
|   | plain washer, 3.2 mm | 100 | reichelt | SKU 3,2 | 0.92 | 0.92 |
|   | isolated copper wire, 0.14 $mm^2$, yellow | 20 m | reichelt | LITZE GE | 1.28 | 1.28 |

*Appendix 1—table 1 continued on next page*

*Appendix 1—table 1 continued*

| # | Electronic parts | Amount/ pieces | Supplier/ producer* | Part number* | Unit cost €* | Total cost €* |
|---|---|---|---|---|---|---|
| | isolated copper wire, 0.14 mm², red | 20 m | reichelt | LITZE RT | 1.28 | 1.28 |
| | isolated copper wire, 0.14 mm², black | 20 m | reichelt | LITZE SW | 1.31 | 1.31 |
| | USB-A to USB-B 2.0 cable | 5 m | reichelt | DELOCK 83896 | 3.31 | 3.31 |

*The reported part numbers and prices are from German suppliers in 2017. Equipment necessary for soldering, drilling holes and making threads are not listed.

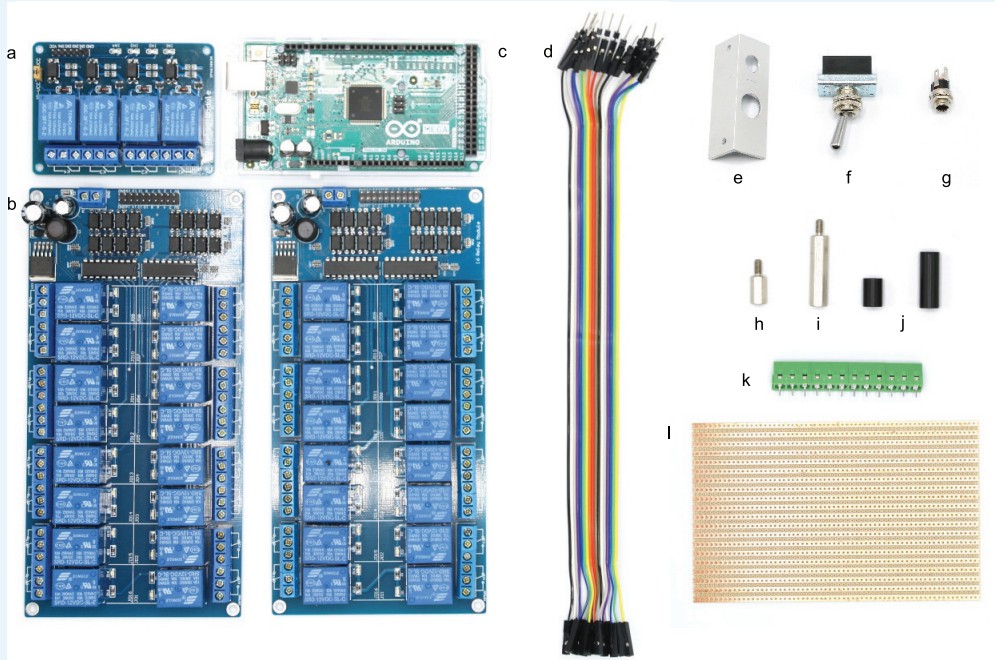

**Appendix 1—figure 1.** Picture of electronic parts with labelling.

**Appendix 1—table 2.** Table of pneumatic parts referring to *Appendix 1—figure 2* including part number and cost.

| # | Pneumatic parts | Amount | Supplier/ Producer* | Part number* | Unit cost €* | Total cost €* |
|---|---|---|---|---|---|---|
| m | push-in fitting, multiple distributor four outlets, G1/8 external thread, 4 mm tubing | 1 | Esska | IQSQ184G0000 | 3.29 | 3.29 |
| n | push-in reducing connector, 10 mm/8 mm tubing | 1 | Esska | IQSG10800000 | 2.25 | 2.25 |
| o | push-in Y-fitting, 4 mm tubing | 1 | Esska | IQSY40000000 | 1.50 | 1.50 |
| p | push-in fitting, G1/8 internal thread, 8 mm tubing | 1 | Esska | IQSF18800000 | 1.62 | 1.62 |
| q | push-in fitting, G1/8 internal thread, 4 mm tubing | 3 | Esska | IQSF18400000 | 1.64 | 4.92 |
| r | push-in fitting, G1/4 external thread, 4 mm tubing | 6 | Esska | IQSG144G0000 | 1.08 | 6.48 |

*Appendix 1—table 2 continued on next page*

*Appendix 1—table 2 continued*

| # | Pneumatic parts | Amount | Supplier/ Producer* | Part number* | Unit cost €* | Total cost €* |
|---|---|---|---|---|---|---|
| s | push-in fitting, G1/8 external thread, 4 mm tubing | 7 | Esska | IQSG184G0000 | 0.96 | 6.72 |
| t | vacuum ejector | 1 | Esska | 175811211 | 70.26 | 70.26 |
| u | silencer for vacuum ejector | 1 | Esska | SD14MS000000 | 1.06 | 1.06 |
| v | 2/2 solenoid valve | 3 | Esska | 8552MZ12V000 | 32.14 | 96.42 |
| w | pressure regulator, 1–10 bar | 2 | Esska | R018-6000000 | 29.85 | 59.70 |
| x | precision pressure regulator, 10–1000 mbar | 2 | Esska | DRF31GS00000 | 68.38 | 136.76 |
| y | 3/2 solenoid valve, body ported | 12 | SMC | S070C-6BG-32 | 38.04 | 456.48 |
| z | 3/2 solenoid valve, body ported manifold | 20 | SMC | S070M-6BG-32 | 34.06 | 681.20 |
|  | U end plate assembly | 2 | SMC | SS070M01-2A | 14.85 | 29.70 |
|  | d end plate assembly | 2 | SMC | SS070M01-3A | 14.85 | 29.70 |
|  | PE-tubing 50 m, outer diameter 4 mm, inner diameter 2 mm | 1 | Esska | 7031PL4 × 2SCH | 11.72 | 11.72 |
|  | silicone tubing 25 m, outer diameter 4 mm, inner diameter 2 mm | 1 | Roth | 9559.1 | 20.35 | 20.35 |
| * | thread sealing tape | 1 | Esska | 929500064514 | 1.67 | 1.67 |

*Use thread sealing tape to seal the connections between push-in fittings (p,q,r,s) the vacuum ejector (t), the 2/2 solenoid valves (v), the pressure regulators (w, x) and the multiple distributor (m).

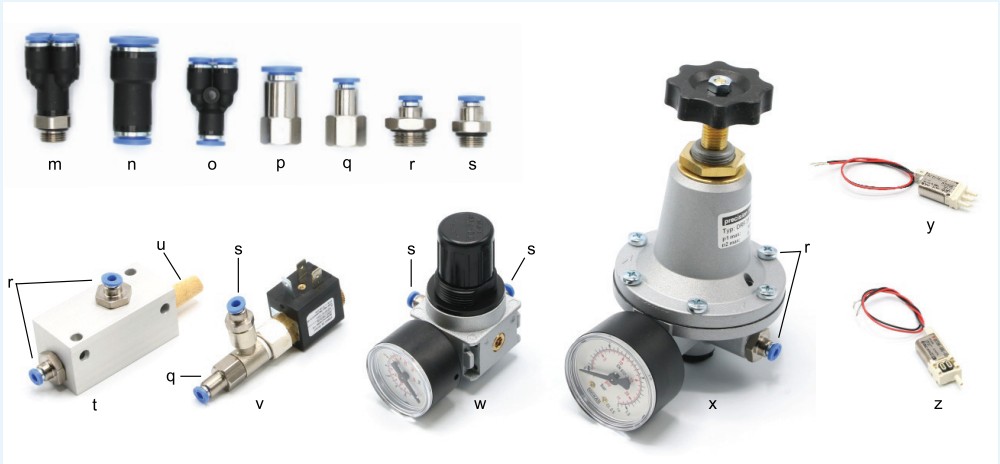

**Appendix 1—figure 2.** Pictures of pneumatic parts with labelling.

## Layout of acrylic sheets

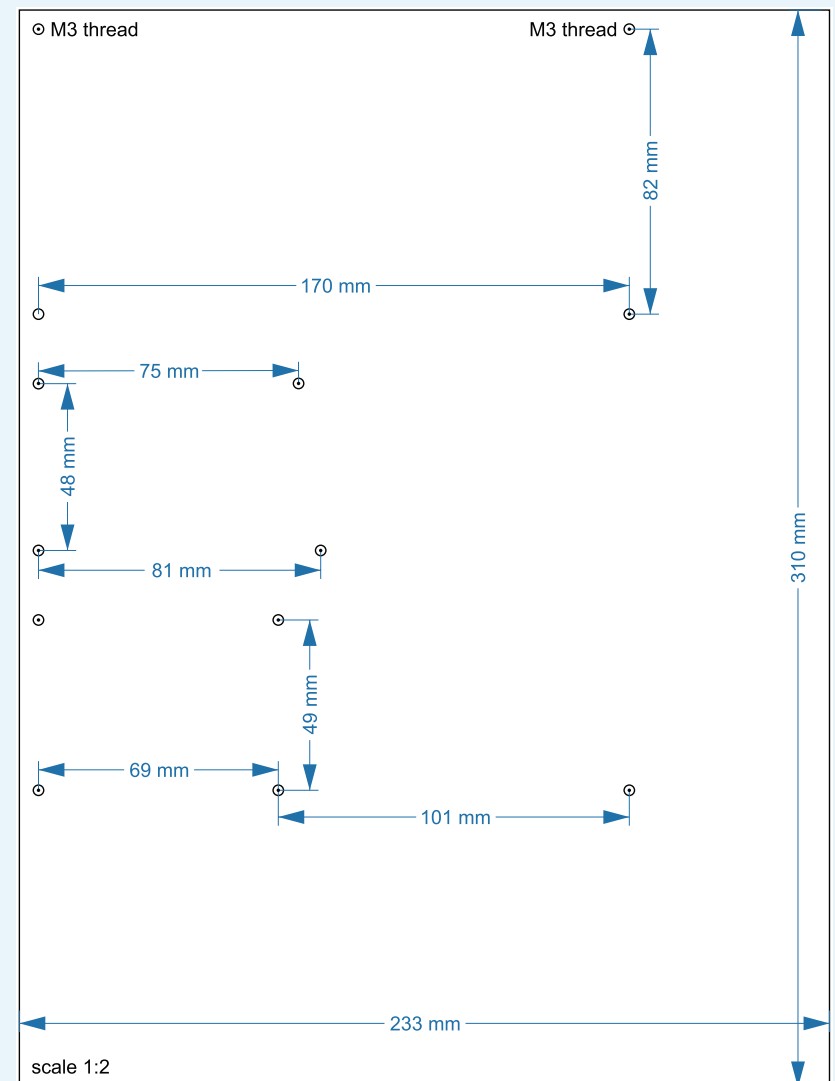

**Appendix 1—figure 3.** Layout of lower acrylic sheet with 10 mm thickness (13). Circles indicate positions where a M3 thread needs to be cut. The scale is 1:2, a 1:1 enlarged layout could serve as a printed template.

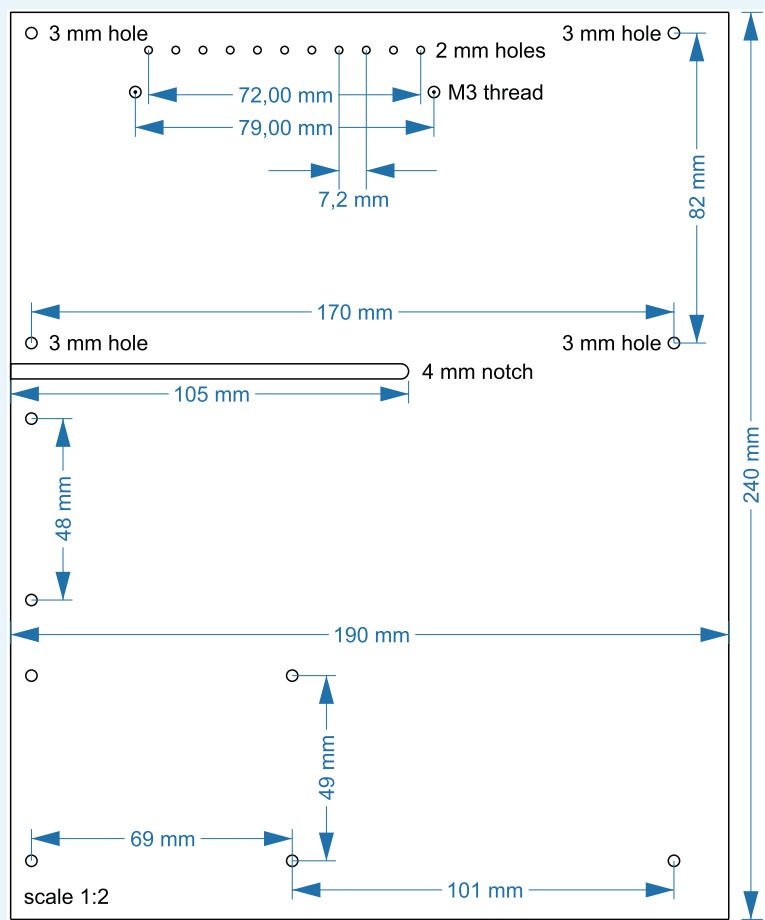

**Appendix 1—figure 4.** Layout of upper acrylic sheet with 5 mm thickness (14). Cut 2x M3 threads at circles with spiral. Drill 11 × 2 mm holes, each 7,2 mm apart. All other circles indicate 3 mm holes. A 4 mm notch will allow for passage of cables. The scale is 1:2, a 1:1 enlarged layout could serve as a printed template.

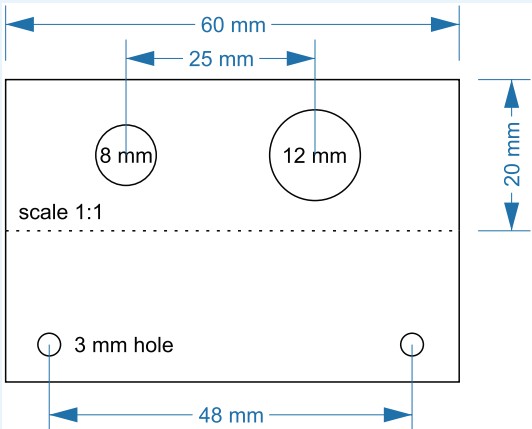

**Appendix 1—figure 5.** Layout for aluminum angle (5). Cut the angle at a length of 60 mm. Drill an 8 mm hole for DC barrel connector (7). Drill a 12 mm hole for toggle switch (6). 3 mm holes are needed to attach the angle to the upper plate.

## Wiring scheme of components

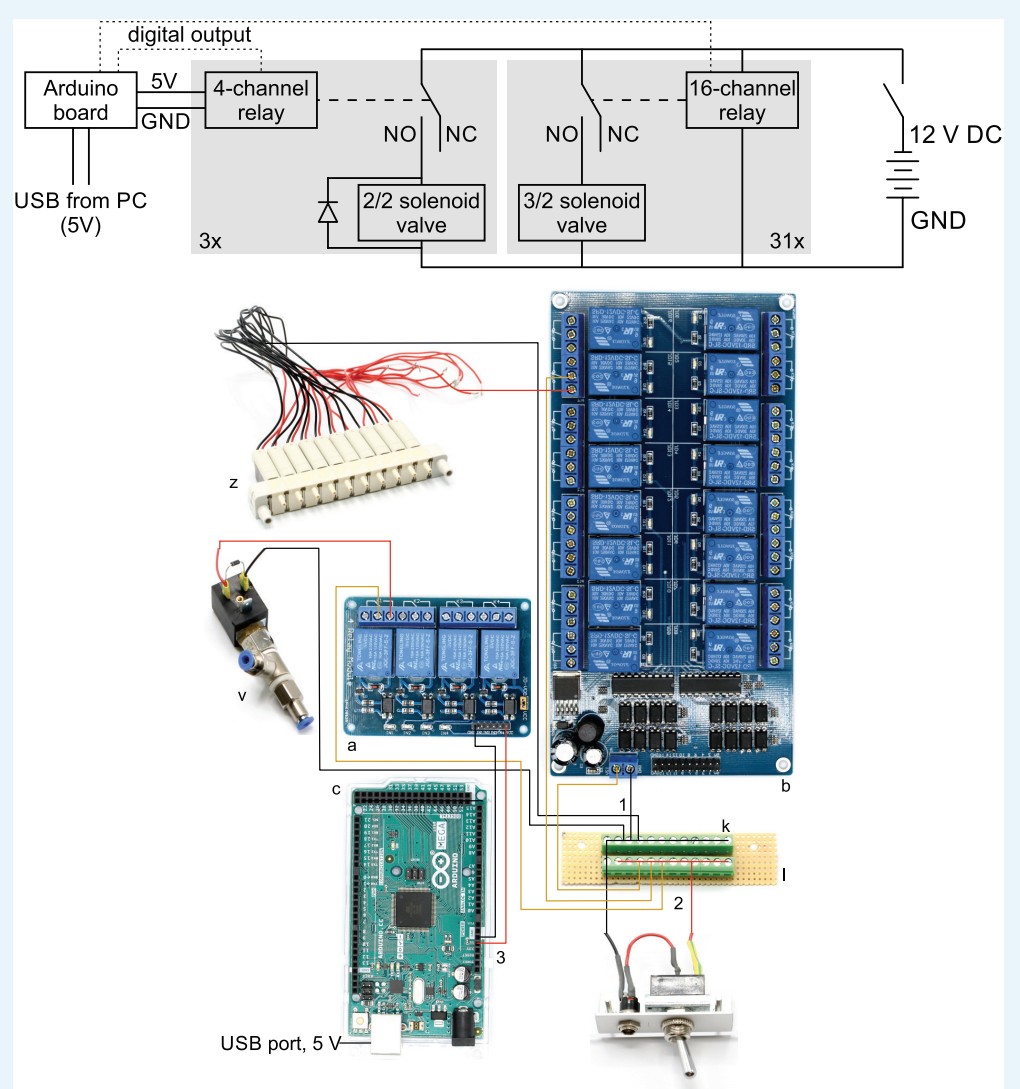

**Appendix 1—figure 6.** Top: Electrical circuit wiring diagram for valves, relays and Arduino board. This wiring diagram shows that all solenoid valves (**z, v**) and the two 16-channel relay board (**b**) are powered through the 12 V DC power supply (80 W, 12 V / 6,67 A). The 4-channel relay is powered through the Arduino board which receives a 5 V power supply from the USB cable. The gray boxes indicate that this circuit is replicated for each valve in parallel. The relays receive digital signals from the Arduino boards (dotted lines) and switch the contacts from NC (normally closed) to NO (normally open) which brings the solenoid valve into a different state. A flyback diode is necessary for the 2/2 solenoid valves to protect from voltage spikes. Bottom: Photographic wiring scheme of circuit. (1) Ground conductor is connected to all black lines through the soldered screw block on the stripboard (**k, l**). (2) The 12 V phase conductor (red) is connected to all yellow lines through the other soldered screw terminal block. These yellow wires carry 12 V to power the 16-channel relay board (**b**), the individual 3/2 solenoid valves (**z**) and the 2/2 solenoid valves (**v**). (3) The 4-channel relay board (**a**) is powered through the 5 V output pin of the Arduino board (**c**) which is supplied through the USB-cable from the PC.

## Assembly guide for pressure control device

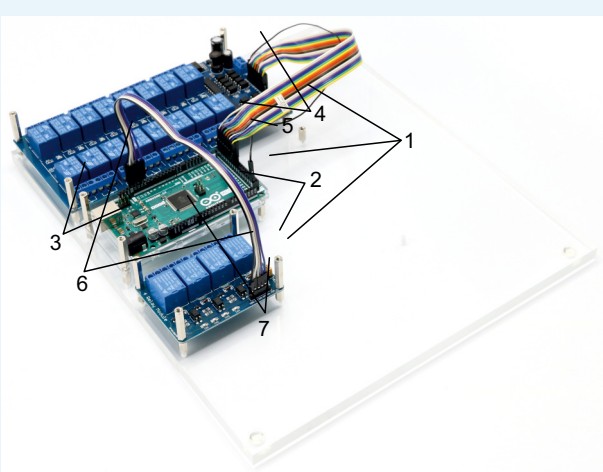

**Appendix 1—figure 7.** Illustration of step 1-7. 0. Prepare the acrylic sheets.
1. Screw in 10 mm metal spacers (h) in M3 threads of the base plate.
2. Place 16-channel relay (b), 4-channel relay (a) and Arduino Mega (c) on the corresponding positions and fix them with the 25 mm metal spacers (i).
3. The Arduino only needs 2 × 25 mm spacers to fix it.
4. Connect the input pins of the 16-channel relay with digital outputs of the Arduino (e.g. 22–37) using the jumper wire (d) and document the corresponding pin numbers.
5. Connect the ground pins of the 16-channel relay and the Arduino.
6. Connect the input pins of the 4-channel relay with digital outputs of the Arduino (e.g. 10–13) and document the corresponding pin numbers. Also connect the ground pins.
7. Connect the 5V power pin of the Arduino with the VCC pin of the 4-channel relay to power it (cable not shown in picture).

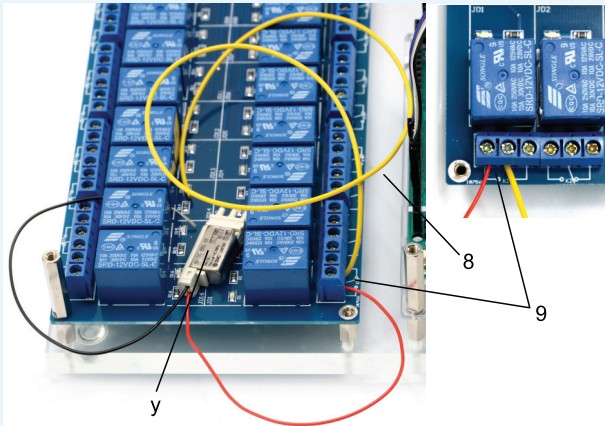

**Appendix 1—figure 8.** Illustration of step 8-9. 8. Cut 31 yellow isolated copper wires at a length of 15 cm each and strip the ends of the cables.
9. Connect the non-manifold 3/2 solenoid valve (y, S070C-6BG-32) to a relay on the 16-channel relay by connecting the red cable to the normally open contact and connecting the yellow cable to the common contact.

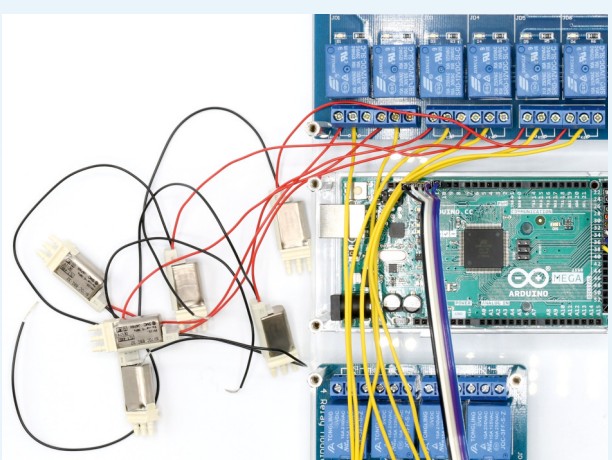

**Appendix 1—figure 9.** Illustration of step 10. 10. Repeat step 9 with all 11 non-manifold 3/2 solenoid valves (y, S070C-6BG-32). Document which relay is connected to each solenoid valve.
This will be important for the programmed control of each valve in the Matlab code.

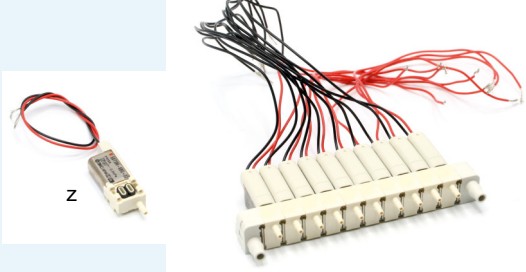

**Appendix 1—figure 10.** Illustration of step 11. 11. Assemble two arrays of 10 manifold 3/2 solenoid valves (z, S070M-6BG-32) as described in the product information.

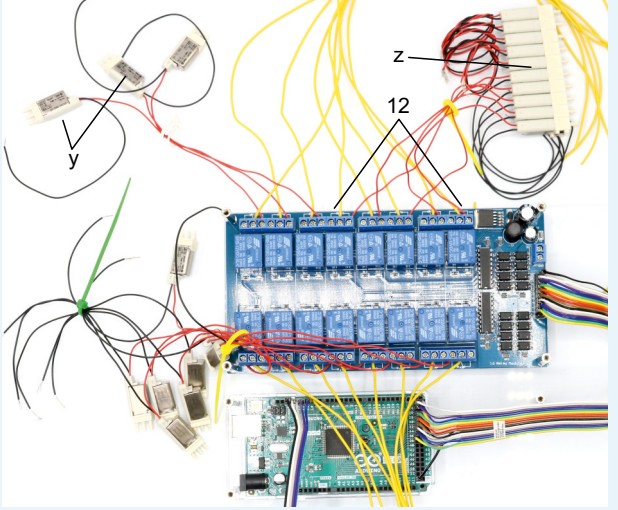

**Appendix 1—figure 11.** Illustration of step 12. 12. Connect 5 manifold 3/2 solenoid valves (z, S070M-6BG-32) to the remaining relays on the 16-channel relay. Connect the red cable to the normally open contact and the yellow cable to the common contact. Document which relay is connected to each solenoid valve.

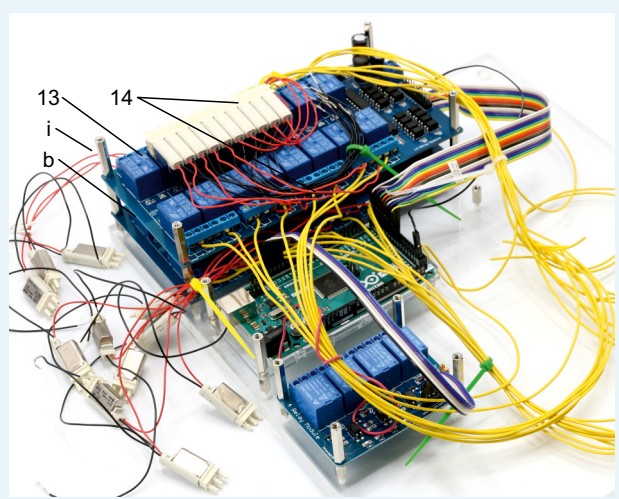

**Appendix 1—figure 12.** Illustration of step 13-14. 13. Attach the second 16-channel relay (b) on top of the other 16-channel relay and fix it with 25 mm spacers (i).
14. Join remaining 5 manifold 3/2 solenoid valves (z, S070M-6BG-32) of the first assembled array with relays of the top 16-channel relay by connecting the red cable to the normally open contact. Connect a yellow cable to the common contact of every relay. Document which relay is connected to each solenoid valve. Pay special attention to order of connections. Intersections of red cables should be avoided. The valve manifold will be oriented as shown in the picture.

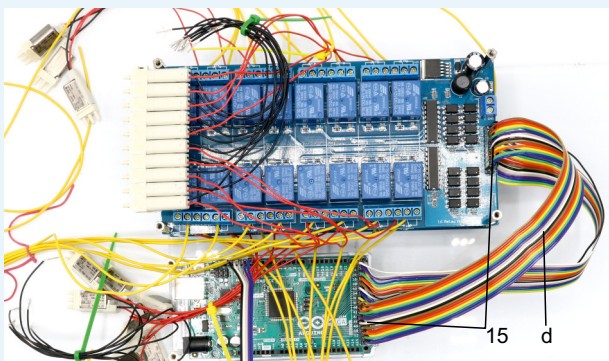

**Appendix 1—figure 13.** Illustration of step 15. 15. Connect the input pins of the top 16-channel relay with digital outputs of the Arduino (e.g. 38–53) using the jumper wire (d) and document the corresponding pin numbers. Connect the ground pins of the 16-channel relay and the Arduino.

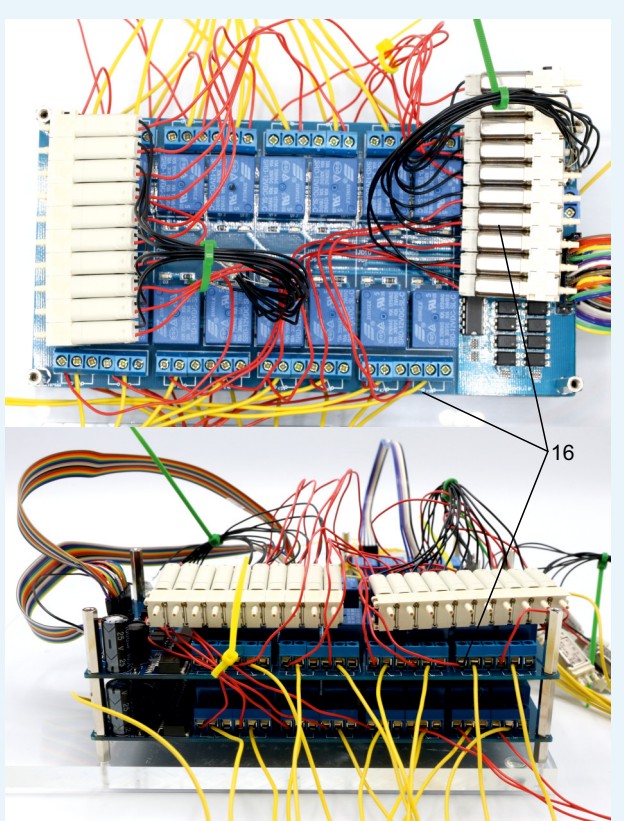

**Appendix 1—figure 14.** Illustration of step 16. 16. Assemble second array of ten manifold 3/2 solenoid valves (z, S070M-6BG-32) and connect these to relays of the top 16-channel relay by connecting the red cable to the normally open contact. Document which relay is connected to each solenoid valve.

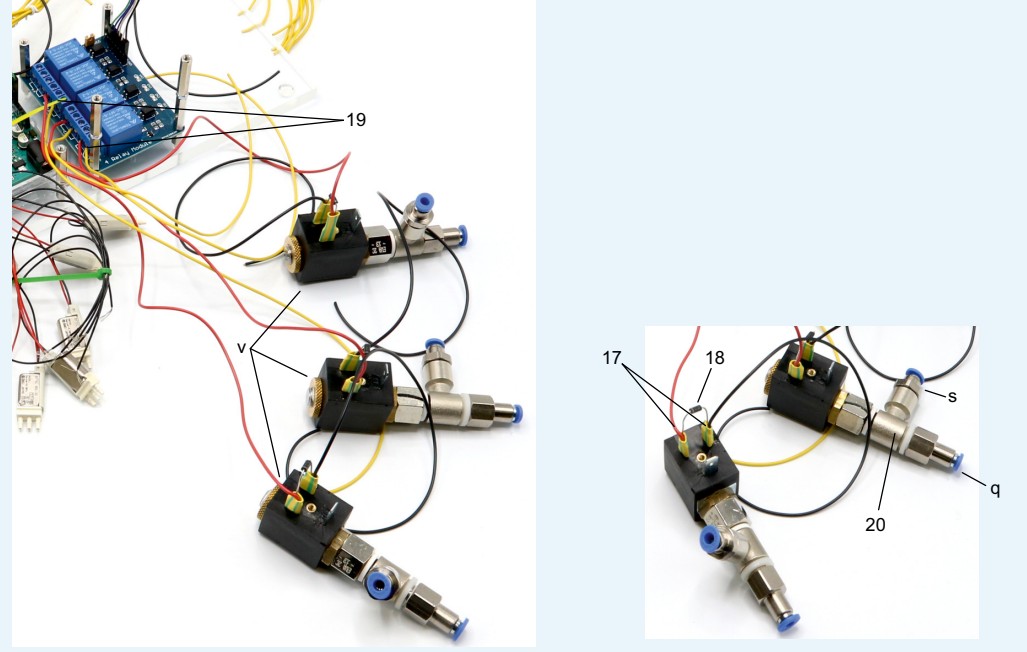

**Appendix 1—figure 15.** Illustration of step 17-20. 17. Solder red and black wires (approx. 10 cm) to contacts of 3 × 2/2 solenoid valves (v).

18. Include a diode (1N4007) between the contacts to avoid current from capacitor discharge damaging the remaining circuit.
19. Connect 2/2 solenoid valves to the 4-channel relay. Connect the red cable to the normally open contact and a yellow cable (approx. 10 cm) to the common contact. Document which relay is connected to each solenoid valve.
20. Screw on push-in fittings (s, q) onto the 2/2 solenoid valves.

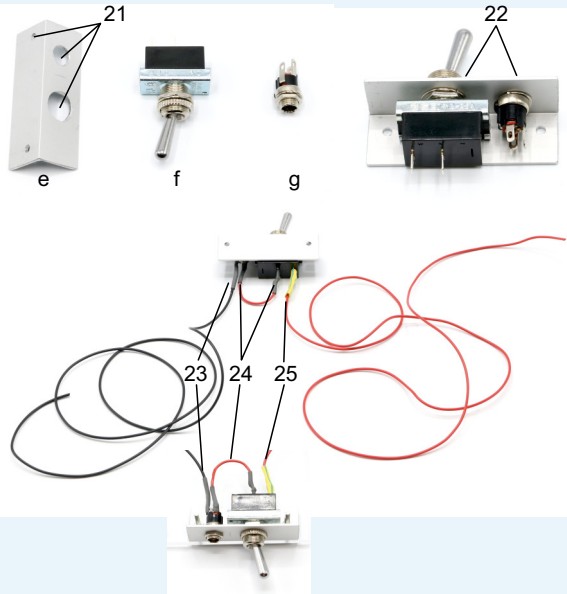

**Appendix 1—figure 16.** Illustration of step 21-25. 21. Drill holes (3, 8, 12 mm) in aluminium angle (e) as shown on the layout.
22. Attach the toggle switch (f) to the 12 mm hole and the DC barrel power connector (g) to the 8 mm hole.
23. Solder a black wire (approx. 10 cm) to the neutral conductor of the power connector. This wire will connect to all ground wires (black wires) of the valves and relays.
24. Solder a red wire to the DC conductor of the power connector and to a contact of the ON/OFF switch.
25. Solder another red wire (approx. 10 cm) to the other contact of the toggle switch. This wire will condcut 12V for all valves and relays (yellow wires).

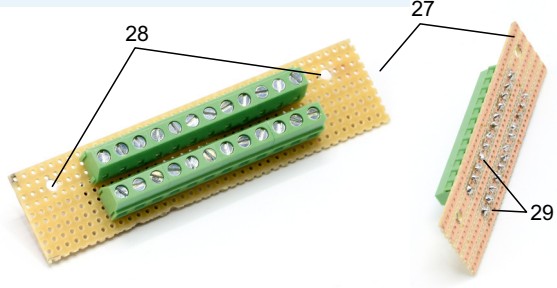

**Appendix 1—figure 17.** Illustration of step 27-29. 27. Cut the stripboard to a 25 × 95 mm rectangle with longitudinal orientation of the conducting striples.
28. Drill two 3 mm holes, 82 mm apart and 7 mm away from the edge. This stripboard will be fixed onto the top acrylic plate above the 16-channel relays.
29. Solder both screw terminal blocks (k) onto the stripboard so that all contacts of one block are connected.

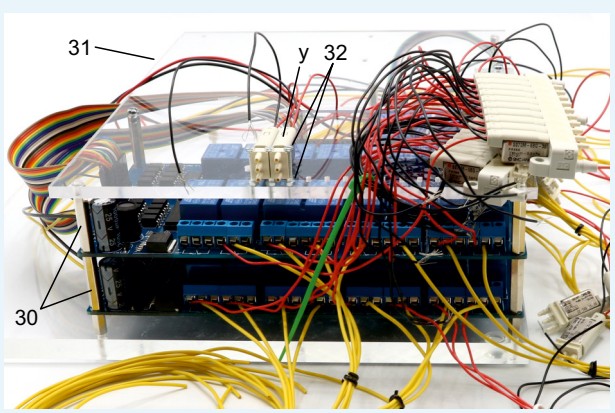

**Appendix 1—figure 18.** Illustration of step 30-32. 30. Screw in second 25 mm metal spacer (i) on top of all metal spacers to create level pillars of metal spacers for the upper acrylic plate.
31. Mount upper acrylic plate (5 mm) on top of the 25 mm metal spacers (i).
32.Use 20 mm M2 bolts to attach the non-manifold 3/2 solenoid valves (y) at the location of the 2 mm holes on the upper acrylic plate. Use M2 nuts below the acrylic plate to fix the valves.

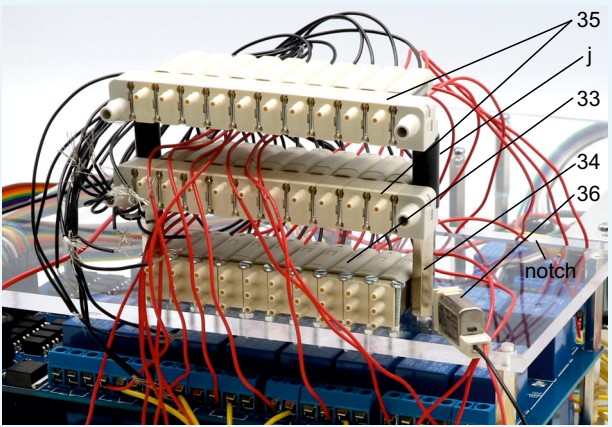

**Appendix 1—figure 19.** Illustration of step 33-36. 33. Repeat step 32 for 10 of the 11 non-manifold 3/2 solenoid valves (z). Document position of each valve, their respective relay and connected Arduino digital output pin.
34. Screw in two 25 mm metal spacers (i) next to the first level of non-manifold solenoid valves.
35. Attach the other two solenoid valve manifolds (z) on top of the metal spacers separated by 20 mm plastic spacers (j) and fixed using 35 mm M3 bolts. Pay attention to avoid twisting of the cables. Route the cables either through the notch or around the upper plate.
36. The 11. non-manifold solenoid valve is later attached on the top plate.

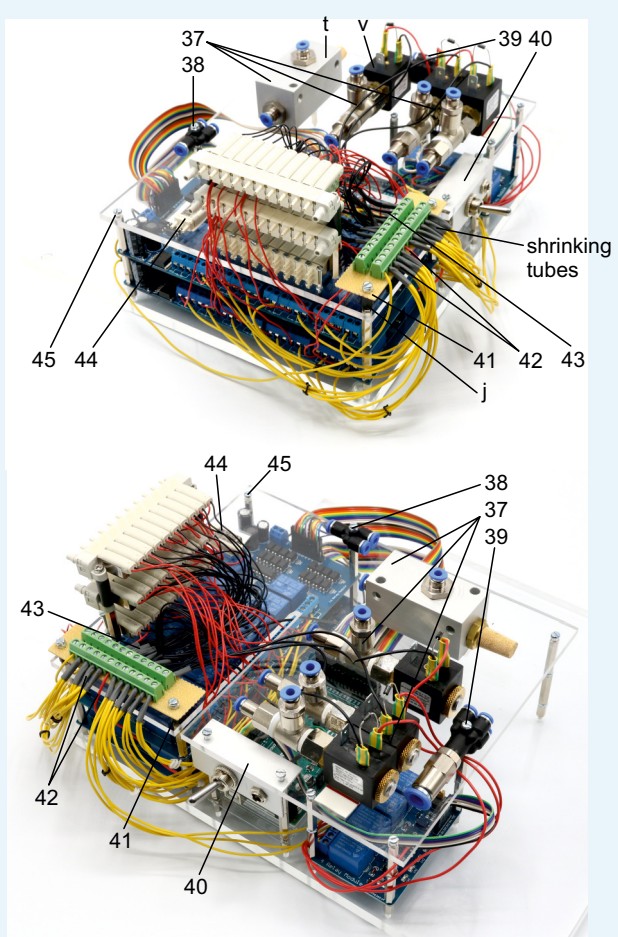

**Appendix 1—figure 20.** Illustration of step 37-45. 37. Attach vacuum ejector (t), equipped with G1/4 push-in fittings (r) and silencer (u), and three 2/2 solenoid valves (v), equipped with G1/8 push-in fittings (s, q), onto the top plate using double-sided adhesive tape.
38. Use 25 mm M3 bolt to attach the push-in Y-fitting (o).
39. Use 35 mm M3 bolt to attach the multiple distributor with four outlets (m, p).
40. Attach the angle with toggle switch and DC power connector (step 21–25) underneath the top acrylic plate, above the Arduino and use M3 bolts to fix it.
41. Attach the cut stripboard with soldered screw terminal blocks (step 27–29) to the top plate using two 25 mm M3 bolts and two 10 mm plastic spacers (j).
42. Connect all yellow wires from the relays and the 16-channel relay boards to one screw terminal block which also connects the red 12V phase conductor from the toggle switch (step 25). This distributes the 12V power from the DC power connector to the relays and thereby to the solenoid valves through the red wires.
43. Connect all black neutral wires from the solenoid valves and the 16-channel relay boards to the other screw terminal block which is also connected with the black neutral conductor wire from the DC power connector (step 23).
44. Use double-sided adhesive tape to attach the 11. non-manifold solenoid valve (step 36) onto the top plate, adjacent to other 3/2 solenoid valves.
45. Use 10 mm M3 bolts on remaining holes.

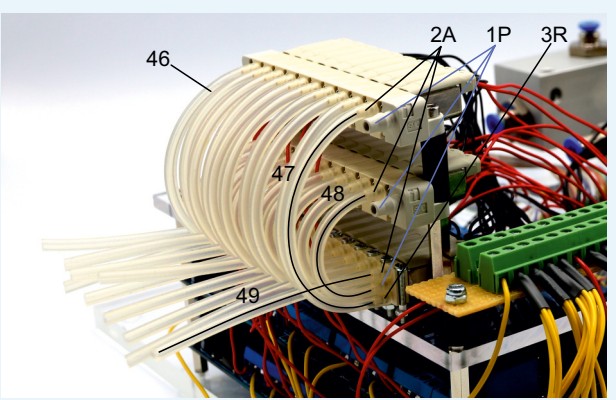

**Appendix 1—figure 21.** Illustration of step 46-49. The following steps connects the 3/2 valves to enable application of different pressures to individual pipette holders.
For schematic depiction, please see respective figure.
46. Cut 10 silicon tubes (diameter 2/4 mm) with a length of 9.5 cm, 10 silicon tubes with a length of 6 cm and 10 silicon tubes with a length of 8 cm.
47. Connect one end of the 9.5 cm tubes to the 2A nozzle of the top level valves and the other end to the lower 3R nozzle of the respective first level valves.
48. Connect one end of the 6 cm silicon tubes to the 2A nozzle of the second level valves and the other end to the middle 1P nozzle of the respective first level valves.
49. Connect one end of the 8 cm tubes to the top 2A outlets of the first level valves. The other end can be connected to tubes that directly connect to the respective pipette holders.

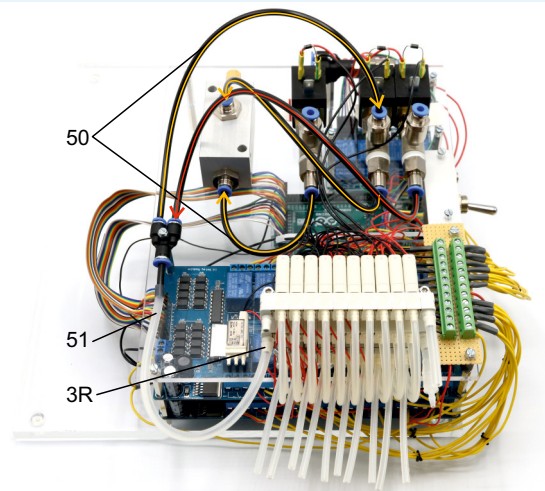

**Appendix 1—figure 22.** Illustration of step 50-51. The following steps connects the 2/2 valves and vacuum ejector to the 3/2 valve manifolds to generate the necessary pressures for the cleaning procedure.
The tubes highlighted in red transfer the 1 bar positive pressure. The tubes highlighted in yellow transfer the negative pressure for suction. Arrowheads indicate direction of air flow.
50. Connect the PE-tubes (outer diameter 4 mm, inner diameter 2 mm) as shown between the 2/2 valves and the vacuum ejector. Connect these to the Y-fitting.
51. Connect the Y-fitting to the 3R nozzle of the second level 3/2 valve manifold through a silicone tube (diameter 4/6 mm).

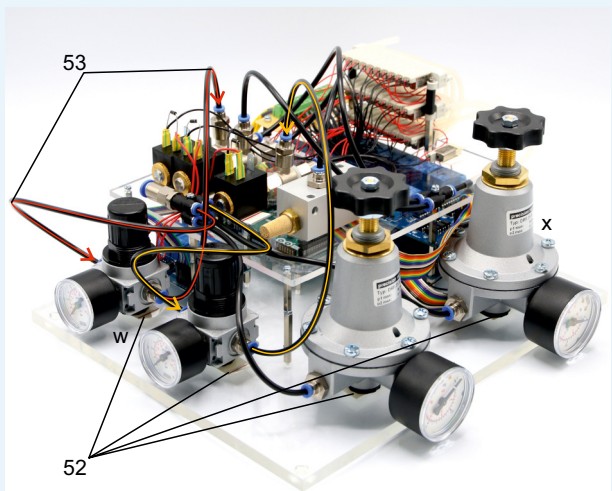

**Appendix 1—figure 23.** Illustration of step 52-43. The following steps connect the 1–10 bar pressure regulators (w) to the pressurized air supply and the 2/2 solenoid valves.
52. Position the pressure regulators (w, x) on the base plate and fix them using double sided adhesive tape. We found this to be of sufficient stability. Pay special attention to the intended direction of air flow depicted on the regulators.
53. Connect the PE-tubes to the multiple distributor with four outlets and to the 1–10 bar pressure regulators (w). Then connect those to the 2/2 solenoid valves as depicted. Meaning of color code and arrowheads correspond to the previous steps.

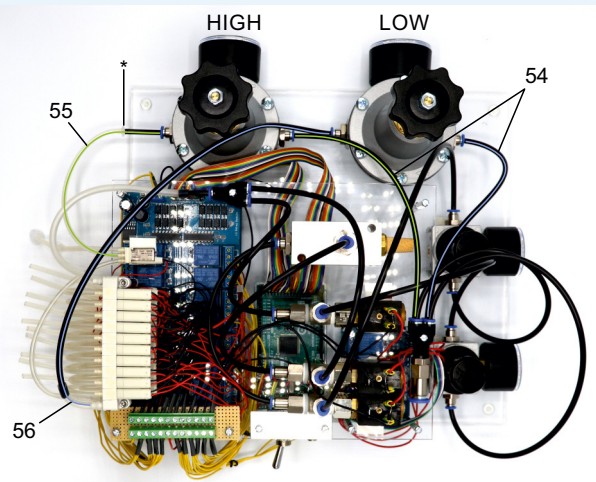

**Appendix 1—figure 24.** Illustration of step 54-56. The following steps connect the precision pressure regulators (x, 10–1000 mbar) to the valve manifold to generate HIGH pressure (70 mbar, green lines) and LOW pressure (20 mbar, blue lines).
54. Connect 2 PE-tubes to the multiple distributor with four outlets and each to one of the precision regulators. Pay spetial attention to intended direction of airflow on the regulators. Arrowheads indicate direction of air flow.
55. Connect the HIGH pressure regulator (set to 70 mbar) to the middle nozzle of the single 3/2 solenoid valve which is attached to the acrylic plate (green line). Connect the PE-tubing with a 2/4 mm silicone tube using a mini tubing connector(*).
56. Connect the LOW pressure regulator (set to 20mbar) to the 1P nozzle of the top level 3/2 valve manifold. Use a 4/6 mm silicone tube to connect the PE-tubing with the nozzle.

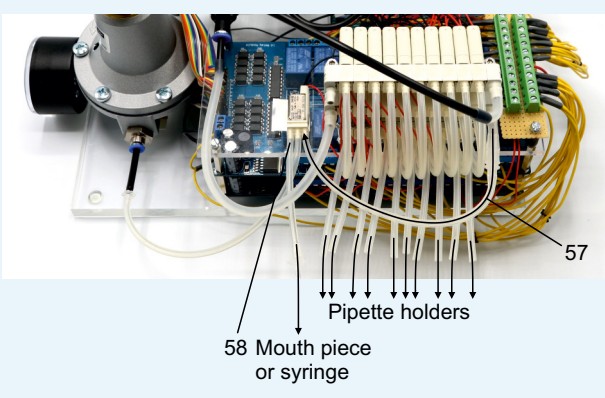

**Appendix 1—figure 25.** Illustration of step 57-58. 57. Connect the 1P nozzle of the second level solenoid valve manifold to the 3R nozzle of the single 3/2 solenoid valve. Use a 4/6 mm silicone tube to connect the 1P nozzle of the manifold. Connect it with a 2/4 mm silicone tube using a mini tubing reducer.
58. Connect a 2/4 mm silicone tube to the 2A nozzle of the single 3/2 solenoid valve which can be connected to a mouth piece or a syringe for applying pressure during membrane sealing and breakthrough.

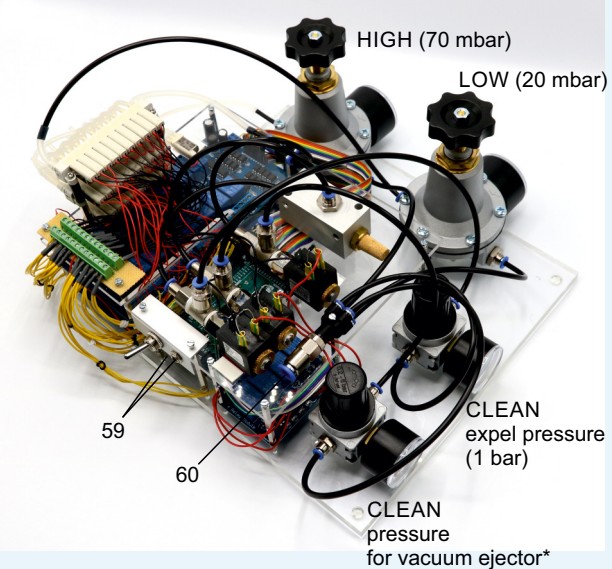

**Appendix 1—figure 26.** Illustration of step 59-60. 59. Plug in the power supply (12 V / 6.67 A, 80 W) and the USB-cable to the Arduino board.
60. Connect to presurrised air supply using a PE-tubing with 8 mm outer diameter. The air supply should be set at sufficient high pressures (e.g. 5 bar).
Adjust the regulators manually to the desired pressures.
*The pressure regulator upstream of the vacuum ejector needs to be adjusted while measuring the negative pressure generated by the vacuum ejector. It should be set at a pressure at which −350 mbar can be applied to the valve manifold (approx. 3–5 bar). To control the device, you can use the Matlab code and GUI we provided. One could also control the Arduino board using any custom script. Before the first use, please transfer the mapping of the individual valves onto their relay number and respective Arduino digital output pin into the script (see guide to Matlab GUI). Now the pressure control device is fully operational.

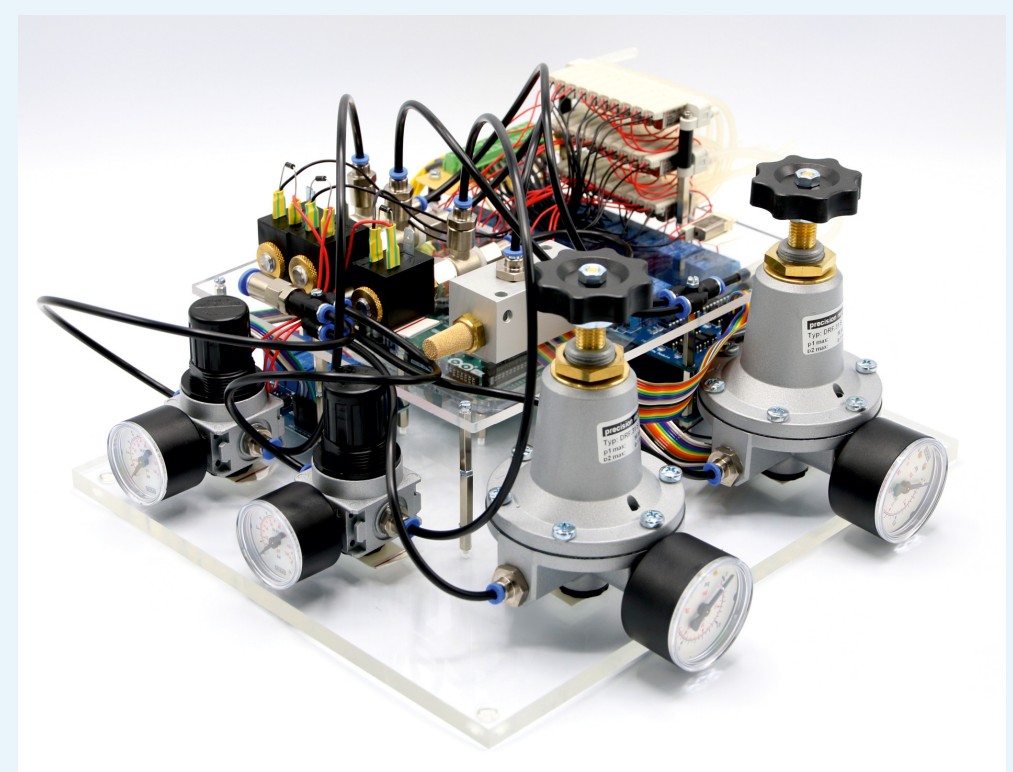

**Appendix 1—figure 27.** Photograph of finished pressure system.

## Appendix 2

# Matlab code

## GUI instructions

We developed a graphical user interface to control the pressure system and the automated pipette movements. The Matlab code for the pressure system can be downloaded from our Github repository: https://github.com/neurocharite/multipatch. For further elaborations regarding the movement algorithms, please see the readme file in the repository.

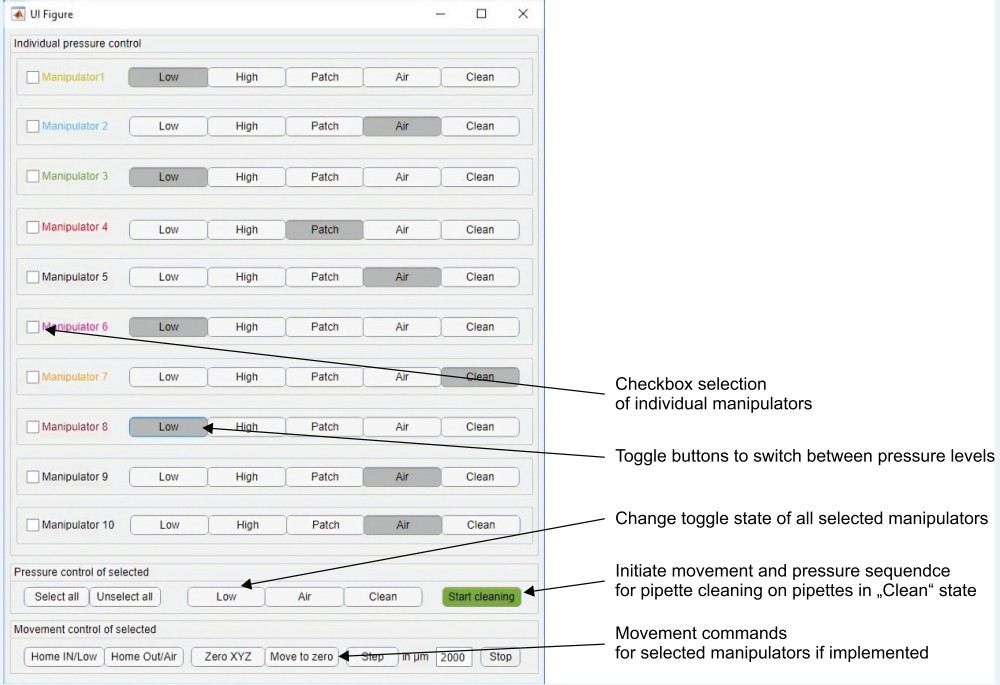

**Appendix 2—figure 1.** Screenshot of MATLAB graphical user interface to control pressure levels and cleaning sequence.

## Pipette positioning algorithm

Manipulator coordinates were matched to the microscope coordinates using a rotation matrix and anchored to a common reference point. Whenever a new region of interest is determined in the microscope, these coordinates are then entered into the pipette positioning program and the pipette tip is moved into the new field of view by calculating the offset. To prevent collision with the slice, we performed this positioning process approximately 2500 µm above the slice. Small manual adjustments are frequently necessary due to varying length of the pipettes. After the pipette tip is manually moved to its dedicated section of the field of view, this new 'IN' position is saved, and the pipette is moved to its initial 'OUT' position. Then the next pipette is moved to the calculated target position and the procedure is repeated until all pipettes were assigned new 'IN' positions. At the end, all pipettes are moved to their assigned 'IN' positions and lowered in the z-axis to roughly 250 µm above the slice surface to prevent effects of the intracellular solution on the slice. Now, individual whole-cell patch-clamp recordings can be established consecutively. This semi-automated approach reduces the risk of breaking the pipette tip and the time needed to complete pipette positioning to 7–9 min.

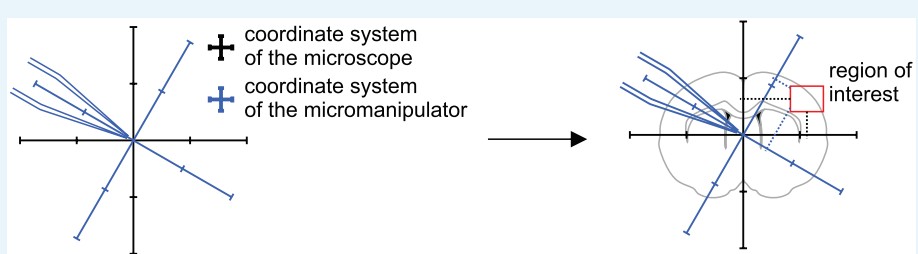

**Appendix 2—figure 2.** Coordinate transformation. Calculate coordinates of micromanipulator corresponding to coordinates of the microscope.

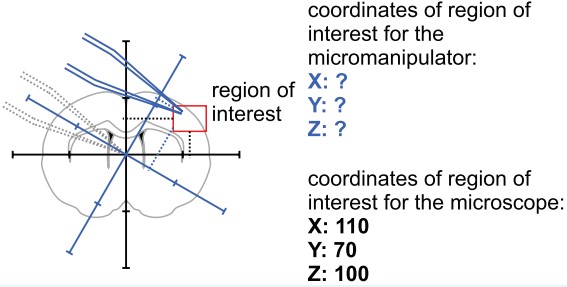

coordinates of region of interest for the micromanipulator:
**X: ?**
**Y: ?**
**Z: ?**

coordinates of region of interest for the microscope:
**X: 110**
**Y: 70**
**Z: 100**

**Appendix 2—figure 3.** Move pipette to calculated coordinate.

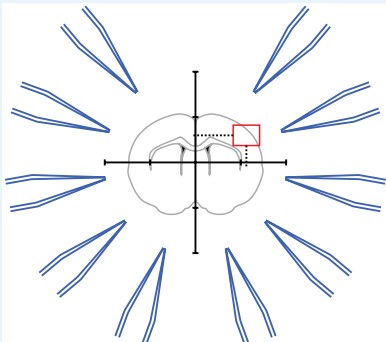

**Appendix 2—figure 4.** Step 1: A brain slice is placed under the microscope. The ‚region of interest ‘is located in the coordinate system of the microscope.

| | |
|---|---|
| X: 110 | X: 50 |
| Y: 70 → | Y: 120 |
| Z: 100 | Z: 100 |

**Appendix 2—figure 5.** Step 2: The coordinates of the ‚region of interest ‘are fed into the pipette finding algorithm. The coordinates for each manipulator are computed.

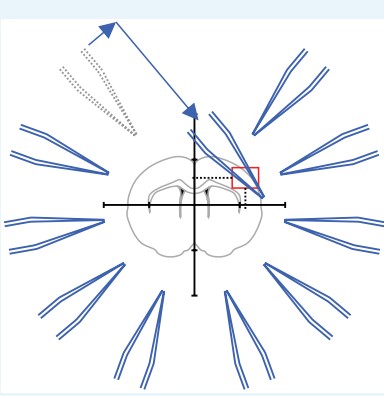

**Appendix 2—figure 6.** Step 3: Pipette one is moved to the calculated coordinate. Manual adjustments are necessary to due variations in pipette shape.

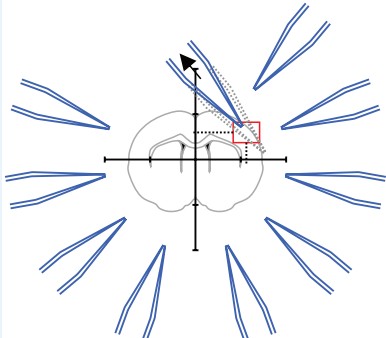

**Appendix 2—figure 7.** Step 4: Pipettes should be moved to a designated corner that will prevent collisions with the other pipettes. This position is then saved.

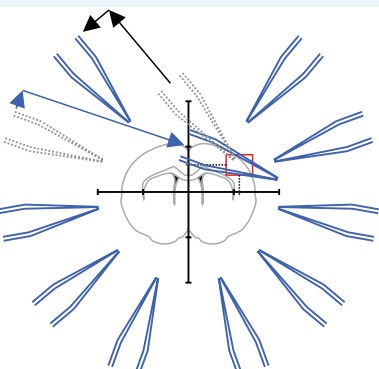

**Appendix 2—figure 8.** Step 5: The pipette is moved to its starting position while the next pipette is moved to the position calculated by the pipette finding algorithm. Repeat step 4 and step five for all manipulators.

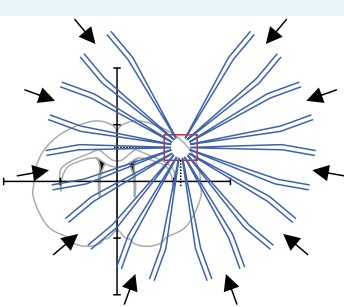

**Appendix 2—figure 9.** Step 6: All pipettes are moved to their target positions simultaneously. All pipttes are now positioned in the region of interest.

## Appendix 3

### Construction drawings

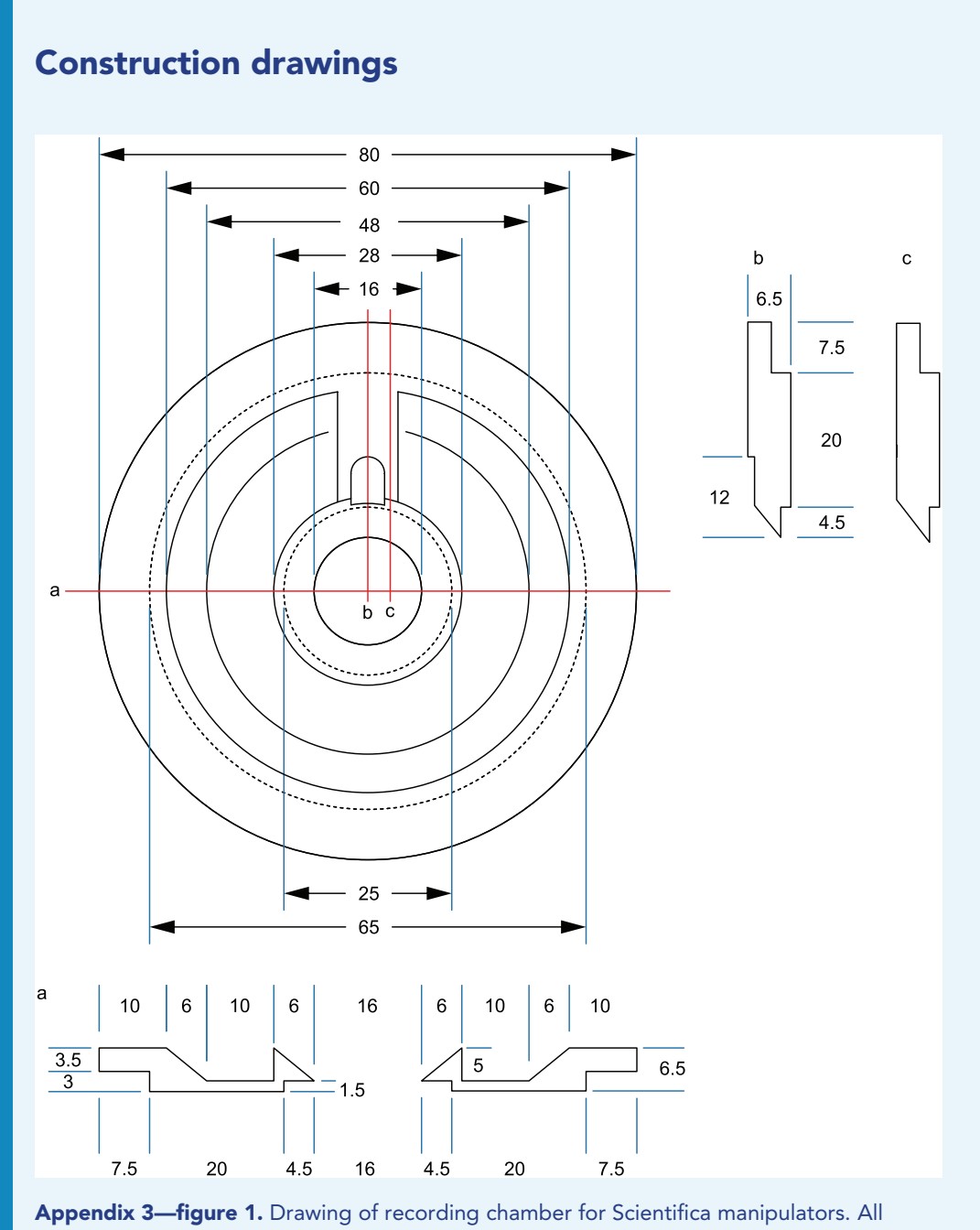

**Appendix 3—figure 1.** Drawing of recording chamber for Scientifica manipulators. All measurements in mm. Dotted lines = edges on underside.

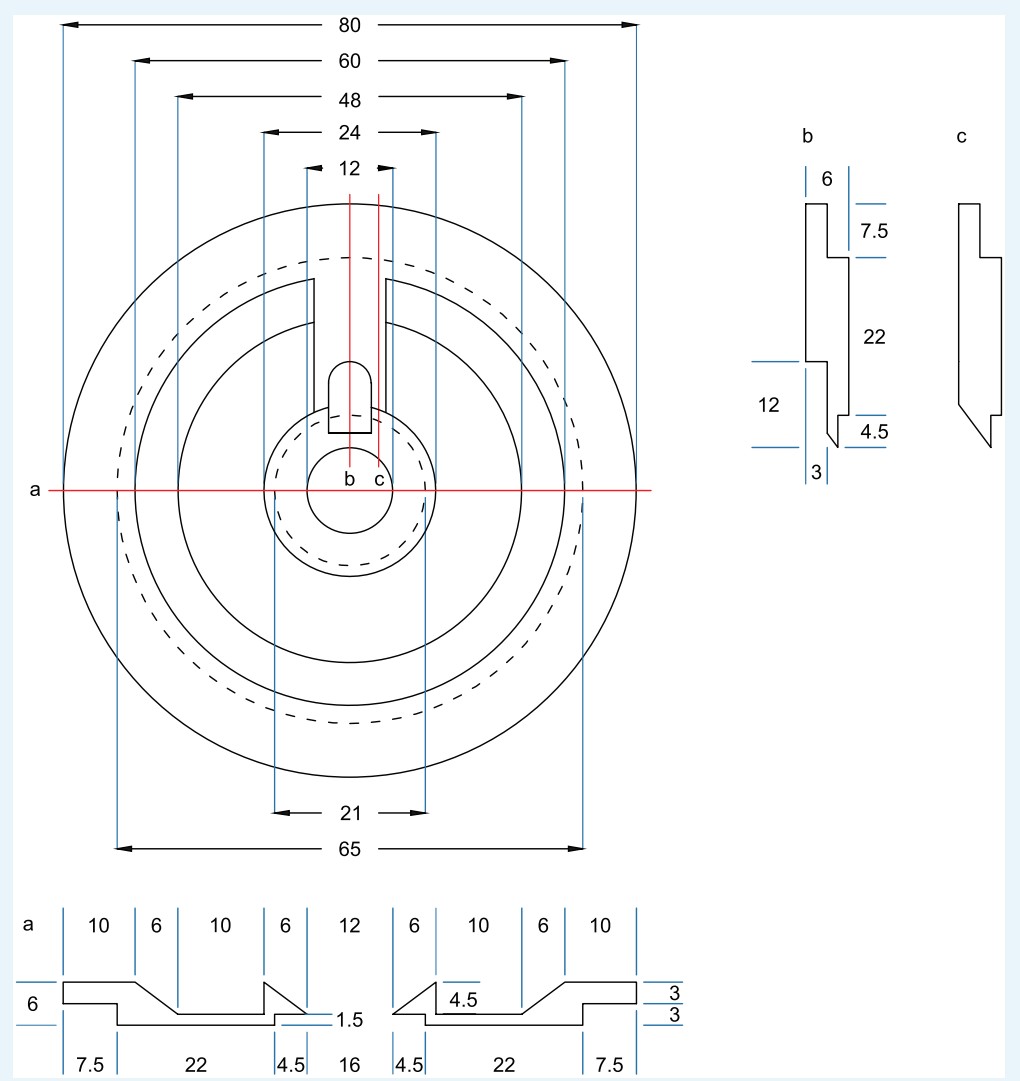

**Appendix 3—figure 2.** Drawing of recording chamber for Sensapex manipulators. All measurements in mm. Dotted lines = edges on underside. Adjustments are needed for Sensapex micromanipulators. Due to a range of motion in the axial axis of 20 mm, the real horizontal range of motion is less than 20 mm. Therefore, the central recording well has a smaller diameter.

## Precautions to prevent contamination of recording chamber with Alconox

At the start of every experiment, a slice is placed into the inner well of the chamber. The outer well remains empty at first. The chamber is then placed into the setup and the inflow and outflow are placed into the inner well for aCSF perfusion. The inlet spans over the outer well in a bridge like manner (A). This prevents wicking of Alconox into the inner well. The drainage has an indentation (C, E) which enables that the outflow maintains a constant aCSF level. That way wicking as well as overflow of aCSF into the outer well are prevented.

Next, the pipettes are mounted onto the headstages. As a last step before starting the experiment, 1–2 ml of Alconox solution is filled into the outer well using a syringe with a long canula or a pipette. Performing these preparatory steps in this sequence minimizes the chance of accidental spill-over. Even though Alconox forms a contact angle <90° with the vertical wall of the outer well, 1 ml of this cleaning detergent leads to a liquid/solid border that is >2 mm away from the top of the wall (D, E) (up to 2.0 ml can be used without the Alconox reaching a

critical level). (D) shows the inner and outer well filled with aCSF and Alconox. The inflow is placed in the aCSF. Black dashed lines indicate the liquid/solid border. There are no structures present that could lead to wicking of Alconox into the aCSF. (B) and (C) are schematic drawings showing the sequence of movements during the beginning of the cleaning protocol. This sequence prevents wicking since the pipette entirely leaves one liquid in a linear motion before it immerges into the other one.

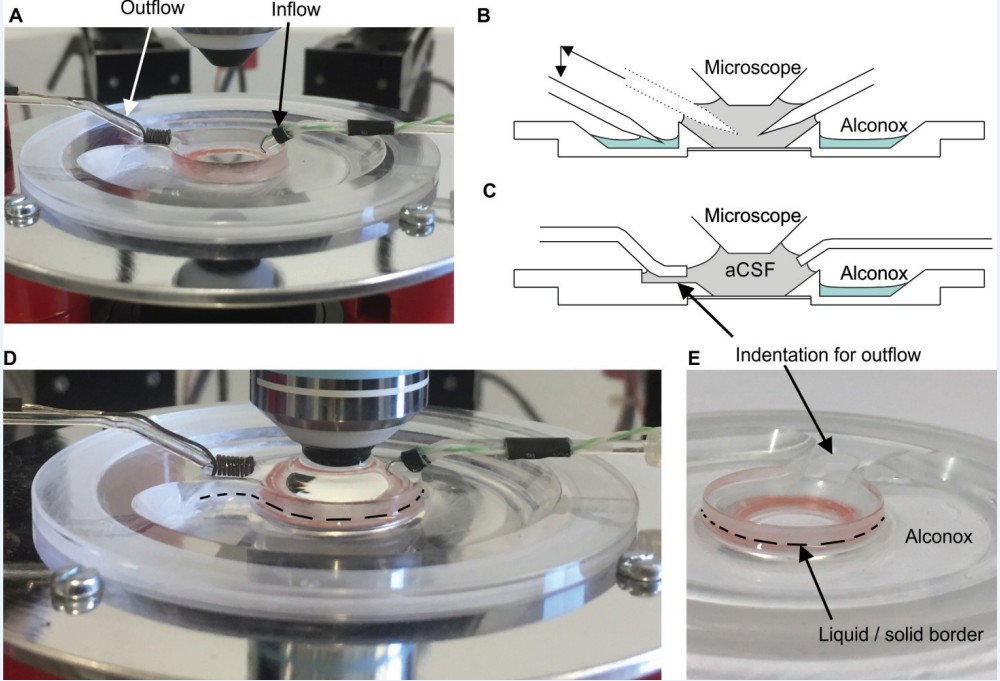

**Appendix 3—figure 3.** Illustration of recording chamber and solutions. (**A**) Photograph of recording chamber with inflow and outflow placed into the central well. (**B**) Schematic view of pipettes in the central well. The left pipette is moved from the center well into the outer circular well containing Alconox. (**C**) Schematic view of in- and outflow. Note that outflow is placed in the indentation which guarantees a constant liquid level in the recording chamber. (**D**) The inflow is placed in the aCSF. Black dashed lines indicate the liquid/solid border. (**E**) Close up view of the liquid/solid border.

## Custom stage
Custom stage for Sensapex micromanipulator with Nikon Eclipse microscope

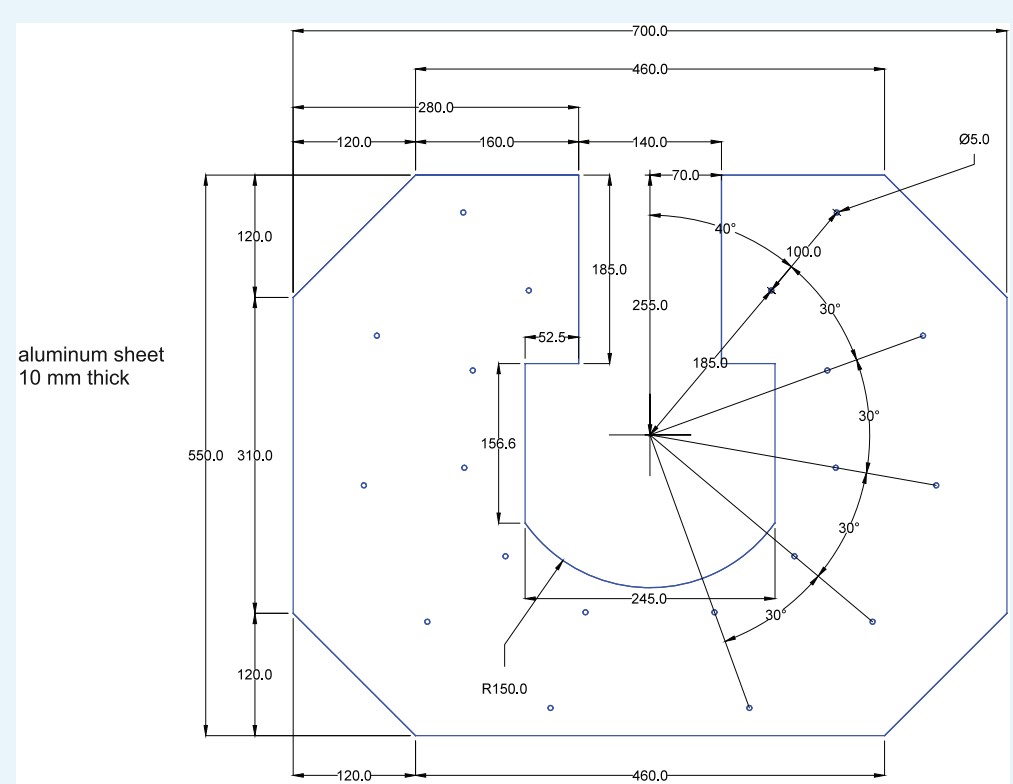

**Appendix 3—figure 4.** 10 mm thick sheet of aluminum. The sheet is shown in blue. Measurements are shown in black. All measurements are in milimeters. All drilling-holes are 5 mm in diameter (for M6 ISO metric screw threads).

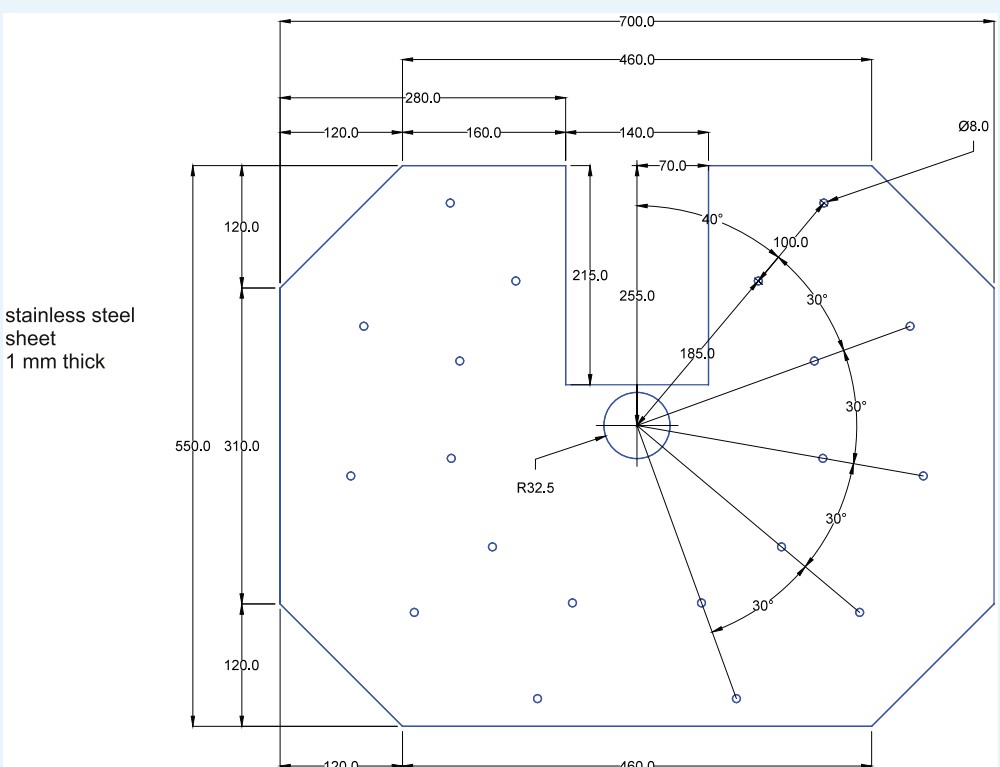

**Appendix 3—figure 5.** 1 mm thick sheet of stainless steel. The sheet is shown in blue. Measurements are shown in black. All measurements are in milimeters. All holes in the

stainless steel sheet are 8 mm in diameter. The two sheets have to be glued together before being mounted into the setup with the stainless steel sheet facing upwards.

## Pipette filling device

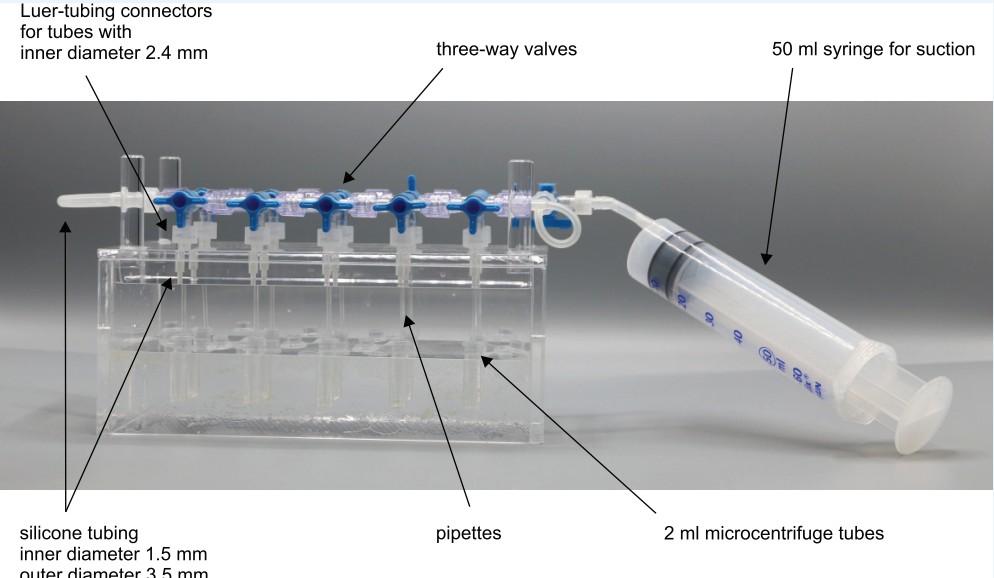

Luer-tubing connectors for tubes with inner diameter 2.4 mm

three-way valves

50 ml syringe for suction

silicone tubing inner diameter 1.5 mm outer diameter 3.5 mm

pipettes

2 ml microcentrifuge tubes

**Appendix 3—figure 6.** Photographic overview of multi-pipette filling device.

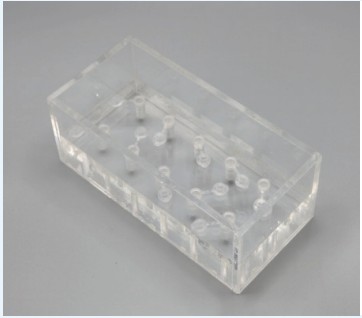
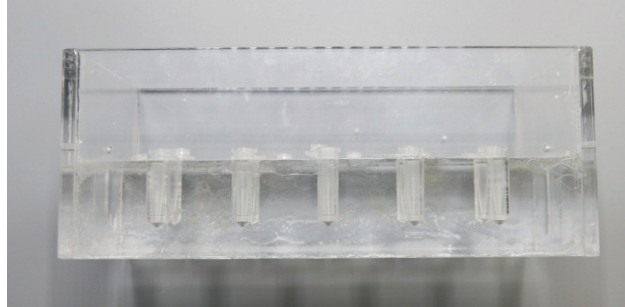

**Appendix 3—figure 7.** Bottom part.

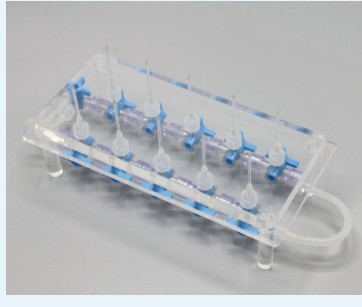
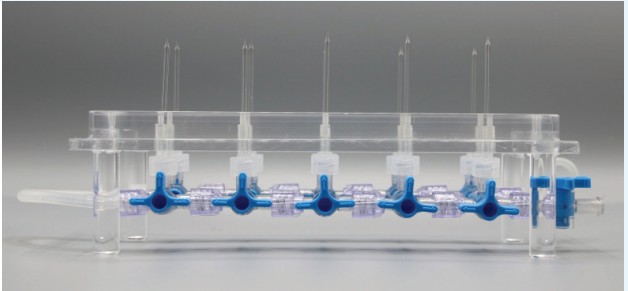

**Appendix 3—figure 8.** Top part.

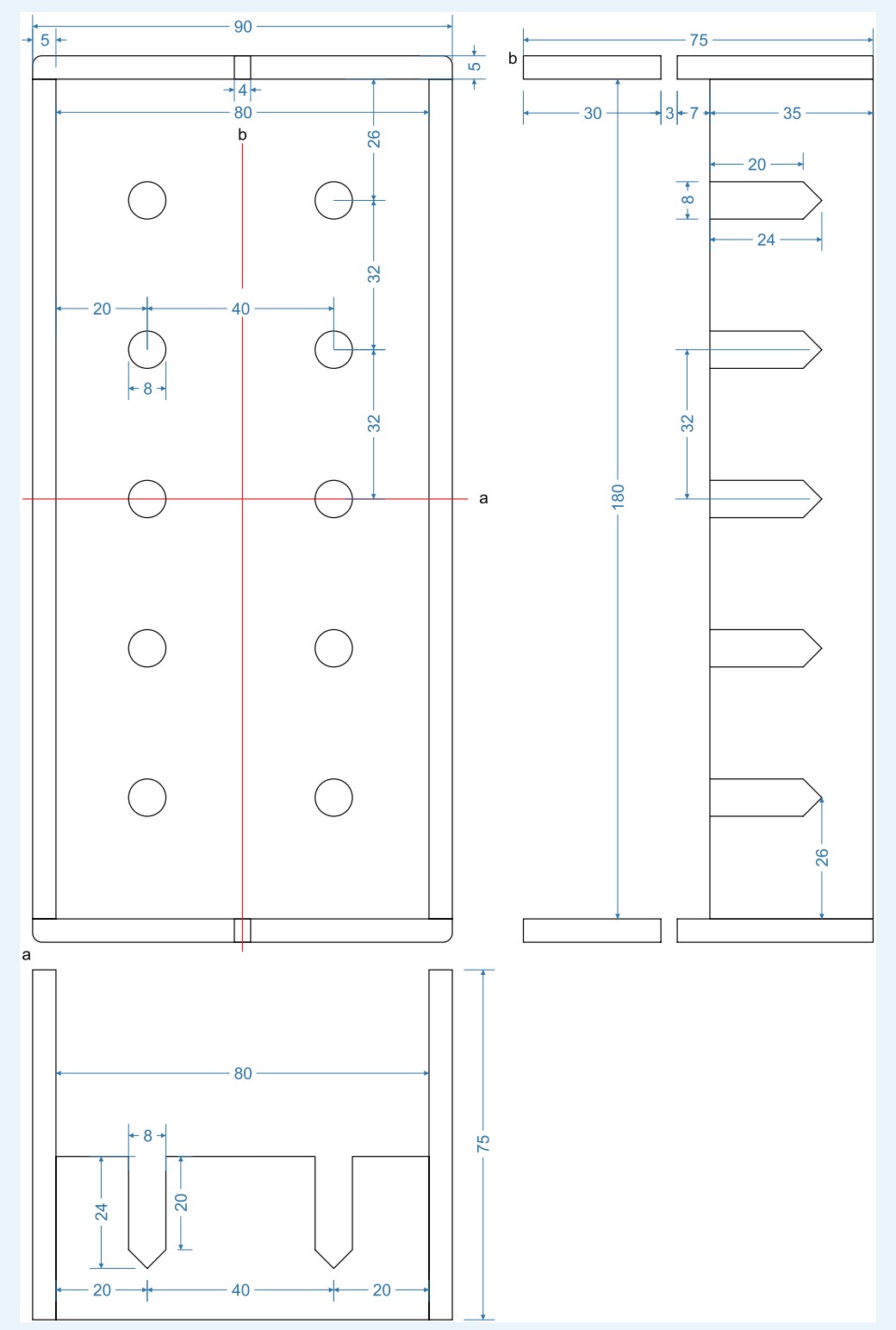

**Appendix 3—figure 9.** Bottom part of pipette filling station. 1,5 ml microcentrifuge tubes with intracellular solution are inserted into the 8 mm holes. Slight adjustment might be necessary to guarantee upright position of the tubes and easy insertion. Distance between the holes (32 mm) are determined by the distance of the central channel between two connected Braun 3-way stopcocks. Slight variation might be necessary when different models are chosen. Height of the walls are determined by the length of the pipettes to ensure that pipettes will reliably

dip into the tubes. The current measurements are optimized for approximately 40 mm long pipettes.

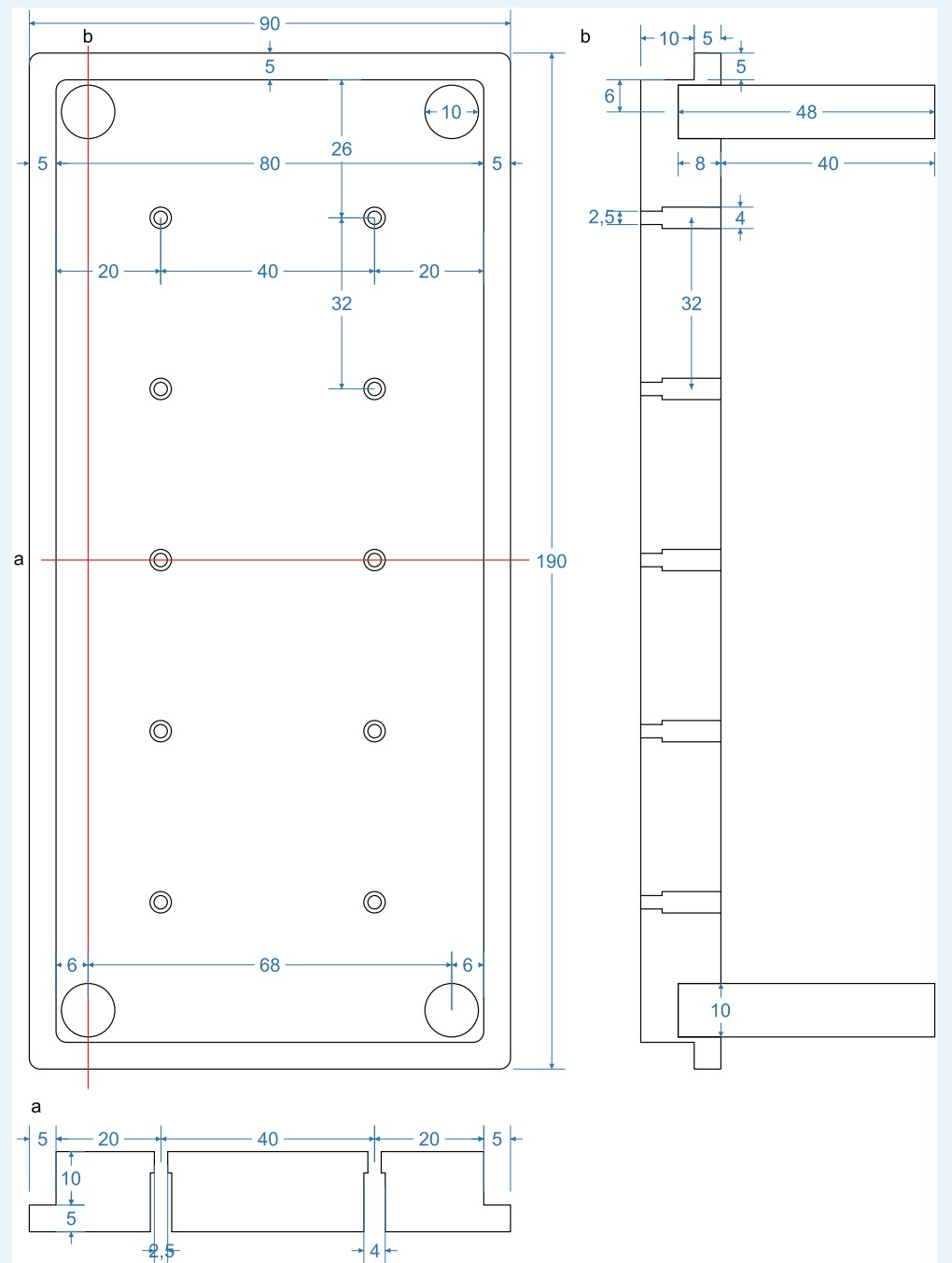

**Appendix 3—figure 10.** Top part of pipette filling station ('lid'). Pipettes will be inserted into the 2,5 mm holes. Silicon tubes with 1,5 mm inner diameter and 3,5 mm outer diameter will be inserted into the 4 mm holes until the narrowing. Section (**a**) depicts the horizontal section through the middle depicting the 2 mm and 4 mm holes. Section (**b**) depicts a longitudinal section through the pillars of the lid. These enable a stable upside-down positioning of the lid in which the pipettes will be pointing upwards. The pillars are glued into a 10 mm hole.

## Analog output routing device

This device is necessary when more headstages are to be used in a multipatch setup than analog output channels are available on the digital-to-analog/analog-to-digital (DAAD-) board.

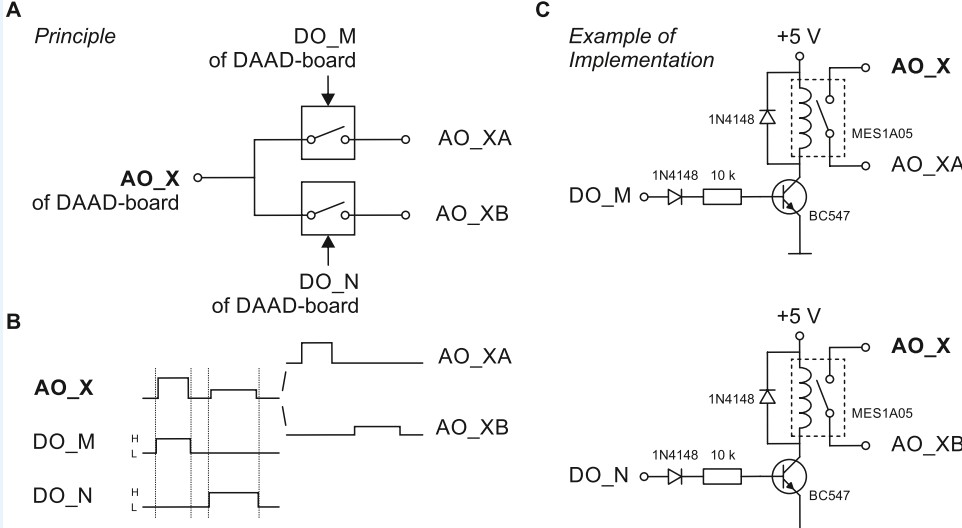

**Appendix 3—figure 11.** Schematic wiring diagram for analog output routing device. (**A**) Principle of a very simple and limited analog-out-switch distributing the analog out signal of the DAAD´s analog output channel AO_X to two channels serving two amplifiers or amplifier channels. The switches are gated by digital output channels of the DAAD-board (DO_M and DO_N). (**B**) Via AO_X two different, non-simultaneous, non-overlapping command signals (different time of occurence, amplitude, duration) are sent to the switch, DO_M and DO_N are set high (**H**) as indicated (note that the digital pulses should start slightly earlier and terminate slightly later than the intended analog signals) and thus gate the respective command to the respective output of the switch (AO_XA or AO_XB). This switch is limited in the sense that the connected AO_XA and AO_XB will carry an identical (voltage- or current-) command signal (technically in both cases a voltage signal) from the DAAD to the individual amplifier inputs if command signals are to be applied simultaneously. (**C**) Example of implementing the principle shown in A. To distribute the analog out command signal of the DAAD-board to two amplifier command input channels, two reed relais (e.g. MES1A05) are used. The power supply can be taken from an USB-port of a computer.

