## [Decision Letter]

**Acceptance summary:**

Peng et al. have adopted and optimized recent innovations such as patch pipette cleaning, multiple amplifiers and automated pressure control to create a very high throughput semi-automated method for simultaneous multiple patch clamp recordings from precious tissue, such as resected brain tissue from epilepsy surgeries. Here they show their robotic system can obtain high numbers of successful paired recordings in part due to a system that allows for cleaning and reusing patch electrodes for multiple serial recordings. This approach yields hundreds of potential neural pairs per patient sample, which will undoubtedly yield novel insights on altered synaptic connectivity relevant to epileptic networks. Detailed plans and control software source code are provided, and the overall approach is validated in both rat and human brain slices. This paper should be a valuable and accessible resource for those needing to optimize the yield of paired intracellular recordings from precious tissue.

**Decision letter after peer review:**

Thank you for submitting your article "High-throughput microcircuit analysis of individual human brains through next-generation multineuron patch-clamp" for consideration by *eLife*. Your article has been reviewed by three peer reviewers, and the evaluation has been overseen by a Reviewing Editor and Eve Marder as the Senior Editor. The following individual involved in review of your submission has agreed to reveal their identity: Tim Jarsky (Reviewer #3).

The reviewers have discussed the reviews with one another and the Reviewing Editor has drafted this decision to help you prepare a revised submission.

Summary:

This Tools and Resource paper describes a low cost, high throughput, multipatching system designed to optimize discovery from precious live brain material, such as that obtained in human epilepsy surgery. The approach utilizes and acknowledges a number of advances developed by other research groups, and the authors provide early results indicating that the approach will highly leverage what can be gained from such precious materials. The methodology is well laid out, with appropriate drawings, photos, parts lists, and videos. It should be an accessible method that would be utilized by exactly those that would most benefit.

Essential revisions:

1) One major concern is the pipette cleaning system the authors adapted. Based on the CR Forest's original protocol, an additional step is needed to clear the residual detergent adhering to the outside surface of the pipette tip with aCSF before moving pipettes into the recording chamber for patch attempt, yet Peng and colleagues omitted this step. There is a good practical reason for omitting this step, as stated by the authors, but little or no data are provided to support this practice. They claim that there were not differences in recording quality or electrophysiological properties between pipettes at first and after cleaning, but these claims should be supported with the data. In particular it would be important to report how the membrane potential, input resistance, synaptic events, and action potential parameters (amplitude, width et al) change over time after patching and repatching.

2) The article overemphasizes connectivity analysis and ignores its limitations, e.g., false negatives. The authors may wish, instead, to emphasize that multipatch recordings are currently the only technique available to assay the strength and short-term plasticity of monosynaptic connectivity.

3) Important missing experimental details include specifying if it is possible to patch cells while recording from other cells, the time that is taken to probe connectivity and an analysis of the distance distribution of recorded cells (e.g., is the distance distribution of cells obtained to extend recordings the same as those obtained initially?).

4) The authors emphasize some benefits of the semi-automated approach but do not identify other aspects of multipatch experiments that could benefit from automation – for example, data acquisition and online QC, and real-time connection detection. With the potential for so much data gathering, data format (e.g. Neurodata without borders), data sharing, automation of connection detection and analysis, should be addressed.

5) While taking advantage of rare live human tissue and especially to increase the data yield from each sample is a valid rationale for developing multi-patching systems, it may it is not known yet whether this will be sufficient yet to probe the difference between individuals. When discussing difference across human individuals, what kind of differences were observed? Cell types, connections? I would suggest the authors to tone down this claim.

[Editors' note: further revisions were requested prior to acceptance, as described below.]

Thank you for resubmitting your work entitled "High-throughput microcircuit analysis of individual human brains through next-generation multineuron patch-clamp" for further consideration by *eLife*. Your revised article has been evaluated by Eve Marder as the Senior Editor, and a Reviewing Editor.

The manuscript has been very much improved but there is one remaining issue that need to be addressed before acceptance, as outlined below:

Regarding the new experiments on repatching following Alconox treatment without Alconox rinsing (subsection “The final expulsion sequence does not require additional wells containing aCSF”). These are convincing and show that in young rat brain tissue that in general there is little cumulative effect of the cleaning process on subsequent neuronal health. There are two points that need clarification here.

1) Do the authors expect that the results with mature human brain tissue will be equivalent to that with young rat brain tissue? Is there any limitation we should be aware of in this validation experiment.

2) Were the same neurons ever repatched following cleaning? This would allow for direct comparison of neuronal properties before and after patch pipette cleaning.

In addition, since each patch electrode was used more than once, arguably the relevant statistical comparison here is repeated measures ANOVA, rather than group population statistics, as appears to have been reported in Figure 3J, K and L.

---

## [Author Response]

Essential revisions:1) One major concern is the pipette cleaning system the authors adapted. Based on the CR Forest's original protocol, an additional step is needed to clear the residual detergent adhering to the outside surface of the pipette tip with aCSF before moving pipettes into the recording chamber for patch attempt, yet Peng and colleagues omitted this step. There is a good practical reason for omitting this step, as stated by the authors, but little or no data are provided to support this practice. They claim that there were not differences in recording quality or electrophysiological properties between pipettes at first and after cleaning, but these claims should be supported with the data. In particular it would be important to report how the membrane potential, input resistance, synaptic events, and action potential parameters (amplitude, width et al) change over time after patching and repatching.

Thank you for acknowledging our rationale for omitting the additional cleaning step. We agree that this practice and our claim for unimpaired physiology of the neurons should be supported by further data. To assess possible effects of our cleaning protocol on recording quality and electrophysiological properties, we have performed additional experiments in acute brain slices from rat motor cortex. We recorded 81 neurons in 12 brain slices from 2 animals (P21, P22) using 28 pipettes with up to 2 sequential cleaning rounds and compared the cellular and synaptic physiology. We excluded 4 interneurons, furthermore, 9 cells with depolarized membrane potential were excluded which were recorded on fresh and cleaned pipettes with equal probability (3/28 cells with fresh pipettes, 2/28 cells after first cleaning and 4/28 cells after second cleaning), please see the Materials and methods section for details.

We plotted the distribution of the cellular and synaptic properties of fresh and cleaned pipettes in Figure 3. Furthermore, we calculated the mean relative change and its confidence interval for all parameters requested. We found that the relative change of the mean value for these parameters were within 10%. We also calculated the confidence interval of these relative mean changes which represents the boundaries of statistically significant equivalence. Overall, we found no evidence for a systematic effect of our cleaning approach on the cellular or synaptic physiology. We have included the statistical results as source data to Figure 3 and adapted the Materials and methods section accordingly.

2) The article overemphasizes connectivity analysis and ignores its limitations, e.g., false negatives. The authors may wish, instead, to emphasize that multipatch recordings are currently the only technique available to assay the strength and short-term plasticity of monosynaptic connectivity.

We thank the reviewer for raising this point and agree that there are limitations regarding connectivity analysis using multipatch recordings. We included a paragraph addressing potential causes for false negatives in the Discussion.

We also agree that synaptic strength and short-term plasticity are important parameters of these connections. While a paired patch-clamp recording configuration represents an optimal approach to analyse these parameters, they can also be determined by combining patch-clamp recordings with 2-photon glutamate uncaging or optogenetic stimulation. However, the presynaptic stimulation reliability might be lower than with the patch-clamp method. We have emphasized the importance of these parameters and the technical advantage of the multipatch approach in the respective Discussion section.

3) Important missing experimental details include specifying if it is possible to patch cells while recording from other cells, the time that is taken to probe connectivity and an analysis of the distance distribution of recorded cells (e.g., is the distance distribution of cells obtained to extend recordings the same as those obtained initially?).

While probing connectivity simultaneous to patching would save time, we refrained from doing so due to several practical reasons which we outlined in a novel paragraph in the Results section. We also included the time needed to probe the connectivity and measure the intrinsic properties of the cells (Results section).

We thank the reviewer for bringing up the important point that intersomatic distances might influence connection probability and that locations of repatched cells should be controlled to prevent possible bias. As suggested, we have analysed the effect of cleaning-to-extend on the intersomatic distance in our previous rat presubiculum experiments and found a similar distribution between clusters obtained with and without clean-to-extend. We have plotted the distance distributions in Figure 5—figure supplement 1 and discussed them in the Results section.

4) The authors emphasize some benefits of the semi-automated approach but do not identify other aspects of multipatch experiments that could benefit from automation – for example, data acquisition and online QC, and real-time connection detection. With the potential for so much data gathering, data format (e.g. Neurodata without borders), data sharing, automation of connection detection and analysis, should be addressed.

We agree that there are multiple aspects of the experiments that can be further automated, and we are also convinced that increased data yield necessitates standardization of analysis and data format. However, we do see a trade-off between experimental flexibility and automation in data acquisition and analysis. Since we wanted to maximize the applicability for other groups and their specific questions in this report, we used commercially available acquisition software while automated trace analysis for connection detection is certainly important and an ongoing effort. While the Signal software could also perform online analysis, we believe that this is only necessary for closed-loop experiments. We included a novel paragraph on these issues in the Discussion section. We further support the effort of open science and development of standardized data format to facilitate collaborations. We have provided suggestions on this topic in the Discussion section.

5) While taking advantage of rare live human tissue and especially to increase the data yield from each sample is a valid rationale for developing multi-patching systems, it may it is not known yet whether this will be sufficient yet to probe the difference between individuals. When discussing difference across human individuals, what kind of differences were observed? Cell types, connections? I would suggest the authors to tone down this claim.

We understand the concerns of the reviewer regarding the statistical power of our sample sizes to detect meaningful differences between individuals. We want to emphasize that our main goal of obtaining large samples from single patients is not to determine these differences between single individuals but rather to gain the capacity to assess inter-individual variability. We believe that this is crucial since the tissue is obtained from a very heterogenous group of patients. Analysing the data at the individual level can help us identify invariant parameters which could point towards common principles of the human cortex. On the other hand, parameters with high inter-individual variability should be analysed with caution and subject to further investigation. Therefore, we believe that obtaining statistically meaningful datasets in single patients is an important step to motivate and guide future studies. We have rephrased and detailed our claim in the manuscript to better reflect this aspect (Abstract; Introduction; Discussion).

In our preliminary analysis, we did not find significant differences in the pyramidal cell interconnectivity between patients, while we determined that the confidence interval of differences in connection probability were in the range of -5% and 9.5%. Overall, we believe that a full analysis and discussion of potential invariant and variant parameters are beyond the scope of this technical report and are better addressed in a separate research article. We have added the statistical analysis in the respective Results and Materials and methods section.

[Editors' note: further revisions were requested prior to acceptance, as described below.]

The manuscript has been very much improved but there is one remaining issue that need to be addressed before acceptance, as outlined below:Regarding the new experiments on repatching following Alconox treatment without Alconox rinsing (subsection “The final expulsion sequence does not require additional wells containing aCSF”). These are convincing and show that in young rat brain tissue that in general there is little cumulative effect of the cleaning process on subsequent neuronal health. There are two points that need clarification here.1) Do the authors expect that the results with mature human brain tissue will be equivalent to that with young rat brain tissue? Is there any limitation we should be aware of in this validation experiment.2) Were the same neurons ever repatched following cleaning? This would allow for direct comparison of neuronal properties before and after patch pipette cleaning.

We performed the initial revision experiments in rats, because we rarely have human tissue and did not receive any during the revision time. We also did not attempt to repatch the same neurons. However, we were lucky and received human tissue twice in the last two weeks and have now performed additional experiments to assess the effect of cleaning on the electrophysiological properties of human neurons.

We have compared properties of neurons either patched with fresh (n = 24) or with cleaned pipettes (n = 9, Figure 3—figure supplement 1). We have also repatched the same neurons with the same cleaned pipette (n = 9, Figure 3—figure supplement 2) or another fresh pipette (n = 5, Figure 3—figure supplement 3). We could show that the intrinsic electrophysiological properties of human neurons were and remained similar across conditions (statistical data and tests in Figure 3—source data 1). While we did see a significant decrease in input resistance in cells repatched with a cleaned pipette, multiple additional factors could contribute to the variability in these neurons, such as the effect of repatching itself or time passed. These cleaning-independent changes are reflected in the variability also found in neurons repatched with fresh pipettes (Figure 3—figure supplement 3).

As we have shown that the pipette itself has no systematic effect on the intrinsic properties, we have also addressed the possibility that the extracellular solution, in which the pipettes were rinsed, could have an effect (Figure 3—figure supplement 4). We have therefore patched clusters of neurons (n = 19) and compared their properties and synaptic connections (n = 7) before and after the cleaning of other pipettes to simulate changes in the extracellular solution after the rinsing. Again, we found that the resting membrane potential and action potential kinetics remained very stable (mean relative difference within 2%) while the input and access resistance increased. We also did not see a specific trend in the postsynaptic amplitudes between these two conditions which showed both slight increase and decrease (n = 7, Figure 3—figure supplement 5, Figure 3—source data 2). We have adjusted the Results and Materials and methods sections in the manuscript accordingly.

Overall, we could show that the results of our cleaning experiments in human neurons are similar to those we have shown in rat neurons, even when the same cell was repatched. We did see that the input resistance decreased in cells repatched with a cleaned pipette and increased in cells recorded in ACSF after rinsing. While repatching, recording time and neuronal variability could affect these parameters, we cannot exclude an effect of pipette cleaning in this case. We therefore emphasize that these validation experiments are limited to our setup and research questions and that any implementation of our cleaning procedure by others should be thoroughly tested for the parameters and in the model organism of interest. Especially, as the adhering detergent on the pipette can depend on multiple factors that need to be addressed and fine-tuned for each experimental setting (subsection “The final expulsion sequence does not necessarily require additional wells containing aCSF”).

In addition, since each patch electrode was used more than once, arguably the relevant statistical comparison here is repeated measures ANOVA, rather than group population statistics, as appears to have been reported in Figure 3J, K and L.

Thank you for this helpful advice. We have performed repeated measures ANOVA for 14 pipettes with which three successful recordings of pyramidal cells were established (fresh, 1x cleaning, 2x cleaning). It showed no significant trend and we included the results in Figure 3—source data 2. We also adapted the Results and Materials and methods sections accordingly.